

# The Oceanographic Multipurpose Software Environment

Inti Pelupessy[1,2], Ben van Werkhoven[3], Arjen van Elteren[2], Jan Viebahn[1],
Adam Candy[4], Simon Portegies Zwart[2], and Henk Dijkstra[1]

[1]Institute for Marine and Atmospheric Research Utrecht, Utrecht University, The Netherlands.
[2]Leiden Observatory, Leiden University, The Netherlands
[3]The Netherlands eScience Center, The Netherlands
[4]Civil Engineering and Geosciences, Delft Technical University, The Netherlands

*Correspondence to:* Inti Pelupessy (f.i.pelupessy@uu.nl)

**Abstract.**

In this paper we present the Oceanographic Multipurpose Software Environment (`OMUSE`). This framework aims to provide a homogeneous environment for existing or newly developed numerical ocean simulation codes, simplifying their use and deployment. In this way, `OMUSE` facilitates the design of numerical experiments that combine ocean models representing different physics or spanning different ranges of physical scales. Rapid development of simulation models is made possible through the creation of simple high-level scripts, with the low-level core part of the abstraction designed to deploy these simulations efficiently on heterogeneous high performance computing resources. Cross-verification of simulation models with different codes and numerical methods is facilitated by the unified interface that `OMUSE` provides. Reproducibility in numerical experiments is fostered by allowing complex numerical experiments to be expressed in portable scripts that conform to a common `OMUSE` interface. Here, we present the design of `OMUSE` as well as the modules and model components currently included, which range from a simple conceptual quasi-geostrophic solver, to the global circulation model `POP`. We discuss the types of the couplings that can be implemented using `OMUSE` and present example applications, that demonstrate the efficient and relatively straightforward model initialisation and coupling within `OMUSE`. These also include the concurrent use of data analysis tools on a running model. We also give examples of multi-scale and multi-physics simulations by embedding a regional ocean model into a global ocean model, and in coupling a surface wave propagation model with a coastal circulation model.

## 1 Introduction

Models of the global open ocean have now reached a mature state. Different models, such as `MITgcm`, `MPI-OM`, `POP`, `MOM`, `NEMO`, which have been developed in large international collaborations, are widely used in the community. These models constitute the ocean components in the current CMIP5-



type global climate models, with a horizontal resolution as fine as 25 km, focusing on projected

forecasts of future climate change (IPCC, 2013). They are also used in an ocean-only model configuration (Maltrud et al., 2010) at even higher resolutions (down to about 10 km) to adequately resolve western boundary currents, such as the Gulf Stream, the Agulhas Current and Kuroshio, and to explicitly represent meso-scale eddies.

At the coastal zone, very different models are required, incorporating, for example, tides, river

run-off, sediment transport and wave dynamics (e.g. Zijlema, 2010). In many cases, unstructured mesh models are used (Danilov, 2013; Leuttich and Westerink, 2004) in order to provide an accurate representation (Candy et al., 2014) of complex and irregular domain bounds that strongly influence local flows. An additional challenge in regional coastal ocean models, such as `ADCIRC` and `SWAN`, is that they are not bounded entirely by a coastline and typically contain at least one boundary open

to the global ocean. These open ocean boundaries are usually handled with restoring functions that relax to observations (climatology or transient over a specific period in the past).

In order to evaluate the human-scale impacts of climate change, for example the effect of sea level rise on coastal erosion (Cazenave, 2004), both the open ocean and coastal zone need to be jointly considered. Increasing temperatures and the changes in wind field can give rise to changes in

ocean currents, which in-turn cause dynamical changes in sea level (Brunnabend et al., 2014). These conditions will affect the wave climate and may lead to changes in erosion at sandy coasts. To tackle such problems one can proceed in three ways: nesting of a regional model into a global ocean model (for example, by using the package `AGRIFF`[1]), by developing a model to simulate the physics of both global and coastal flows, or by finding an efficient way to couple two different (e.g., open and

coastal) models together.

In this paper, we follow the latter approach, borrowing from ideas in the astrophysical community. In simulations of the formation of stars and galaxies, a wide variety of codes need to be combined. For example, hydrodynamic codes (describing interstellar gas dynamics) are coupled with N-body codes (for the gravitational dynamics of stars) and processes on different scales, ranging

from planetary to galactic, compete to determine the evolution of the coupled system. Given the need to correctly capture the interactions of the processes represented in the different codes, the community has come up with a Python framework (`AMUSE`) allowing easy interaction of different codes (Portegies Zwart et al., 2013; Pelupessy et al., 2013).

In oceanography similar problems for multi-scale and multi-physics are encountered, and a num-

ber of coupling frameworks exists in the earth system modelling community (e.g. Hill et al., 2004; Buis et al., 2006; Gregersen et al., 2007; Jacob et al., 2005; Larson, 2005; Peckham et al., 2013; Valcke, 2013), These can be roughly divided into (Valcke et al., 2012) *integrated* and *coupling library* approaches, where the former splits codes into elemental units after which the framework merges them into a coupled executable, and the latter approach makes an API available to codes

---

[1] `http://www-ljk.imag.fr/MOISE/AGRIF/`



such that concurrently running codes can share information. The example of AMUSE provides a useful alternative since it takes the approach of integrating different codes in a high-level programming language (Python), using physically motivated programming interfaces to communicate with seperately running instances of the simulation codes. This has the benefit of the parallelism and flexibility provided by a coupling library approach, and the benefit of abstracting much of the bookkeeping in-

herent to code couplings using modern high-level constructs. In this way quite complex simulations can be described in compact scripts, that can be easily understood and easily distributed.

The aim of this paper is to present OMUSE, a framework which adapts the AMUSE approach for use in the ocean modeling community. In section 2, the design and architecture of OMUSE is presented, with a particular focus on data structures, unit conversion and grid remapping. The initial set of

codes included is presented in section 3. In section 4 we discuss the code coupling features of the OMUSE framework with particular emphasis on a quasi-geostrophic model as a conceptual test case. In section 5, we present simple applications of OMUSE showing its capabilities. A summary and discussion of these results concludes the paper (section 6).

## 2 Design and Architecture

As inherited from AMUSE, the basic idea of OMUSE is the abstraction of the functionality of simulation codes (*the community code base*) into physically motivated interfaces that hide their complexity and numerical implementation. OMUSE provides the user optimized building blocks that can be combined to design numerical experiments. The requirement of the high-level glue language is not so much performance, but one of algorithmic flexibility and ease of programming. Hence, a modern

interpreted scripting language with object-oriented features, in our case Python (van Rossum, 1995), is the natural choice. Furthermore, Python has a large user and developer base in scientific computing, and many libraries are available. Amongst these are libraries for numerical computations, data analysis and visualization, which can be used in an OMUSE scripts.

An OMUSE application consists, roughly speaking, of a *user script*, an *interface layer* and the

*community code base* (Pelupessy et al., 2013), as illustrated in Fig. 1. The user script is constructed by the user and defines a numerical experiment by specifying the initial data, the simulation codes to be used and the interactions between the codes. It may include analysis or plotting functions, in addition to writing simulation data to file. The setup and communication with a community code is handled by the framework in the interface layer, which consists of a communication interface

with the community code as well as unit handling facilities and an object-oriented interface. The interface layer also ensures the consistency of the interactions with the various simulation codes by maintaining a state model for each.

Below we give an overview of the design and architecture of OMUSE (as inherited from AMUSE, more details can be found in Pelupessy et al., 2013). The main developments compared with AMUSE



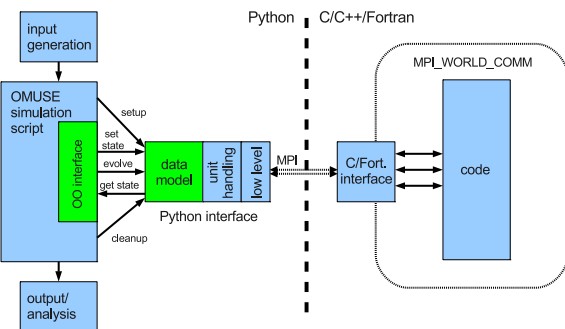

**Figure 1.** Design of the `OMUSE` framework. This schematic representation shows the design of the interface to a community code ("code") and the way it is accessed from the `OMUSE` framework. The code has a thin layer of interface functions in its native language (e.g. Fortran) which communicates through an MPI message channel with the Python host process. On the Python side, the user script ("`OMUSE` simulation script") makes only generic calls to a high-level interface. This high-level interface calls the low-level interface functions, hiding details about units and the code implementation (the communication through the MPI channel does not interfere with the code's own parallelization because the latter has its own MPI_WORLD_COMM context). Adapted from Pelupessy et al. (2013).

```
(1)  qg=QG()
(2)  qg=QG(debugger="gdb")
(3)  pop=POP(number_of_workers=8)
(4)  pop=POP(channel_type="distributed", hostname="Cartesius",
             number_of_workers=600)
```

**Figure 2.** Examples of the instantiation of simulation codes within `OMUSE`. (1) simple instantiation on a local machine of the `QG`, (2) instantiation of a code inside a debugger, (3) local instantiation of an MPI-parallel code (`POP`), (4) instantiation of `POP` on a remote machine for a massively parallel high resolution run through the *distributed* channel (see section 2.2).

are, apart from the addition of oceanographic codes: improvements in grid support, amongst these support for curvilinear grids and extensive framework support for grid remappings and grid generation routines. In addition, a number of domain specific units and utility libraries and support for various file formats, such as NetCDF (Rew and Davis, 1990) output, have been added.

## 2.1    Remote function interface

The interface to a community code is provided by a set of functions, each communicating with the code through a remote function protocol. Currently the default implementation in `OMUSE` of this remote function protocol is based on MPI. A community code is started by the instantiation of an



interface object (Fig. 2), transparent to this. Python provides the possibility of linking Fortran or C/C++ codes directly, however we found that a remote protocol provides two important benefits.

First, it provides for build-in parallelism. The choice for an intrinsically parallel interface is much preferable over an approach where parallelism is added a-posteriori, because unless great care is taken in the design, features can creep in that preclude easy parallelization later on. Secondly, a lot of existing simulation codes are not written in a way that allows for multiple instances. They may, for example, use global variables or assume a single global state. This makes it unwieldy to instantiate

multiple copies of the same code when linking directly. Using remote function interfaces means that the codes run as separate executables, and thus this problem cannot occur (in addition this prevent collisions between incompatible libraries when the codes are built with different compilers).

Within the remote data communication channel, the MPI protocol can be replaced by a different method, two of which are currently available: a channel based on sockets and one based on eStep[2]

technology for distributed computing. At present, the sockets channel is mainly useful for cases were a component process is to be run on one machine. As its name implies, it is based on standard TCP/IP sockets. The distributed channel is described in section 2.2 below. When using the MPI channel, different MPI implementations can be used (e.g. `OpenMPI` or `MPICH`), but not mixed.

The interface works as follows: when an instance of an imported simulation code is made, an

MPI process is spawned as a separate process somewhere in the MPI cluster environment. This process consists of a simple event loop that waits for a message from the Python side. It will make the requested subroutine calls on the basis of the incoming message ID and any additional data that may follow the initial MPI message, and subsequently send the results back (Portegies Zwart et al., 2013). Since there is no direct memory access, the interfaces themselves must be carefully designed

to ensure all necessary information for a given physical domain can be retrieved. Additionally, the communication requirements between processes must not be too demanding. Where this is not the case (e.g. when a strong algorithmic coupling is necessary) a different approach may be more appropriate.

Note that the interface design allows the parallelism of MPI parallel codes to be maintained even

when the communication channel uses MPI (`OMUSE` can be used to run massively parallel codes with thousands of processes). This is guaranteed with the recursive parallelism mechanism in MPI-2. The spawned processes share a standard MPI_WORLD_COMM context, which ensures that an interface can be build around an existing MPI code with minimal adaptation (Fig. 1). Other parallelization paradigms, such as OpenMP, are also supported within `OMUSE`. In practice, for the implementation

of the interface for an MPI code, one has to reckon with similar issues as for the stand-alone MPI application. The socket and distributed channels also accommodate MPI parallel processes. The choice between the different available channels depends on the computing resources needed for a

---

[2]http://estep.esciencecenter.nl





```
(1) q = 1. | units.Sv
    dt= 1. | units.day
(2) (q*dt).as_quantity_in(units.m**3)
(3) (q*dt).value_in(units.km**3)
(4) def Reynolds_number(vel, length, visc):
        return vel*length / visc
(5) R = Reynolds_number( 0.1 | units.cm/units.s, 1000. | units.km,
        1.e-6 units.m**2/units.s)
```

**Figure 3.** An illustration of the use of the `OMUSE` unit algebra module, with (1) definition of a scalar quantity using the | operator, (2) conversion of a quantity to different units, (3) conversion of quantity to float, (4)+(5) definition of a function and its call using quantities.

given run. For runs distributed over remote machines the distributed channel may be required, while locally on a cluster the MPI channel often provides the most optimized communciation path.

## 2.2 Distributed computing

Current computing resources available to researchers are more diverse than simple workstations: clusters, clouds, grids, desktop grids, supercomputers and mobile devices complement stand-alone workstations, and in practice one may want to take advantage of this ecosystem.

To run in such a "Jungle computing environment" (Seinstra et al., 2011), `OMUSE` implements a communication channel based on eStep technology (Drost et al., 2012). This channel starts a daemon and connects with it, to communicate with remote workers. This daemon is aware of local and remote resources and the middleware (e.g. SSH) over which they communicate. The daemon uses the Xenon library to start the worker on a remote machine, executing the necessary authorization, queueing or scheduling automatically. Because `OMUSE` contains large portions of C, C++, and Fortran, and requires a large number of libraries, it is not copied automatically, but it is assumed to be installed on the remote machine. A binary-only release can be generated for resources, such as clouds, that employ virtualization. With these modifications, `OMUSE` is capable of starting remote workers on any computer the user has access to, without significant effort required from the user. From the user point of view, to use the distributed resources, any `OMUSE` script can be distributed by simply adding properties to each worker instantiation in the script, specifying the channel used, as well as the name of the resource, and the number of nodes required for this worker (see Fig. 2).

## 2.3 Unit conversion

In order to simplify the handling of units, a unit algebra module is included in `OMUSE` (Fig. 3). This module wraps standard Python numeric types or Numpy arrays, such that the resulting quantities (i.e. a numeric value together with a unit) can transparently be used as numeric types (see the function definition example in figure 3). Even high-level algorithms, like e.g. ODE solvers, typically do not





```
(1) grid=new_cartesian_grid((100,100))
(2) grid.ssh=0. units.m
(3) subgrid=grid[0:50,0:50]
(4) channel=QG.grid.new_channel_to( grid )
(5) channel.copy_attributes( ["psi"] )
(6) channel.transform( ["ssh"], lambda x:f0/g*x, ["psi"])
```

**Figure 4.** Example usage of the high-level grid data structure: (1) initialization of an empty Cartesian grid, (2) defining an attribute, here a scalar field of sea surface height (3) subgrid generation by indexing, (4) definition of an explicit channel from in-code storage to a grid in memory (5) update of grid attributes over the channel, (6) functional transform over a channel.

need extensive modification to work with OMUSE quantities (and in many cases work without any changes, if they are formulated in a dimensionally consistent way).

OMUSE *enforces* the use of units in the interfaces of the community codes. The specification of
the unit dimensions of the interface functions is part of the interface specification (much in the same way as the data types of the functions). Using the unit-aware interfaces, any data that is exchanged within modules will be automatically converted without additional user input, or - if the units are not commensurate - a code exception is generated. Keeping track of different systems of units and the various conversion factors when using different codes quickly becomes tedious. Enforcing the use
of units therefore eliminates an important source of errors.

### 2.4 Data model

The interfaces to the code send low-level data types (e.g. an array of floats) over the remote function channel. While this is simple and closely matches the underlying C or Fortran interface, it needs considerable duplicated bookkeeping in the user script if used directly. Therefore, in order to sim-
plify working with the codes, a data model is added to the interfaces based on the construction of high-level objects that store the data (Fig. 4). Two base data stores are available: Particle sets and Grids (the main difference between these are that Particle sets can be extended dynamically and are unordered, while Grids are fixed when generated, ordered and can be multidimensional). These data stores can either reference memory in the main Python memory space (for sets defined independent
of any code) or reference the data in the (possibly distributed) memory space of the community code. Subsets can be defined on the sets without additional storage (see fig. 4, these subsets are implemented as views on the underlying local or remote data) and new sets can be constructed using simple operations.

#### 2.4.1 Grid support

Compared to AMUSE, OMUSE expands the support of grid data structures by introducing different grid data types. All types of grids share the same base functionality, including grid sampling and



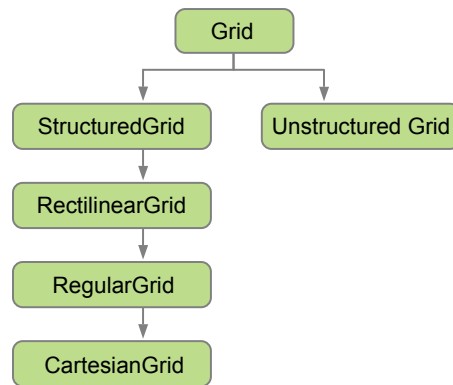

**Figure 5.** Hierarchy of grid data types in OMUSE. Arrows denote inheritance of the corresponding classes in `OMUSE`.

slicing, the creation of save points, and the creation of grid copies that include part or all of the grid attributes. The new grid types form a hierarchy (fig. 5), where each grid type has its own set of (derived) grid attributes (such as cell sizes) and utility functions (for basic operations, such as checking overlap or the extent of a grid). The grid types supported are: Cartesian (single, constant cell size in each dimension), Regular (constant cell size per dimension), Rectilinear (cell boundaries specified per dimension), Structured (cells specified by a grid of corner points) and Unstructured (cell corners are specified for each cell individually).

### 2.4.2 Grid remappings

Grid remapping is a fundamental operation for coupled climate models, where heat and water fluxes are periodically transferred between different component models, each using different grids internally. In many cases, these remappings must be performed in an energy or mass conserving manner to maintain the global conservation conditions of the coupled climate system. As such, `OMUSE` interfaces with `CDO` for their implementation of a second-order conservative remapping scheme (see section 3.2). However, different remapping backends can be used within `OMUSE`.

`OMUSE` extends `AMUSE` with support for remapping quantities between different grids (`AMUSE` included support only for copying data between two equivalent grids). `OMUSE` allows the user to instantiate grid remapping objects. The remapper is initialized by setting the source and destination grid and can be used to remap a list of grid attributes from one grid to the other.

The use of such a remapping object is illustrated in Fig. 6, where as an example, the sea surface height values from one ocean model are remapped to the grid of another ocean model. Note that it does not matter (for the syntax) whether the grid values reside inside the community code or in Python memory. In this example both grids are stored in the memory of the community code, and, if





```
(1) pop = POP(...)
(2) source = pop.elements
(3) adcirc = Adcirc(...)
(4) target = adcirc.elements
(5) remapper = conservative_spherical_remapper(source, target)
(6) remapper.forward_mapping(["ssh"])
```

**Figure 6.** Example usage of the high-level grid remapping functionality in `OMUSE`. In this example, the grid attribute `ssh` (for 'sea surface height') is remapped from the source grid to the target grid, both stored inside the community codes, using a second-order conservative remapping scheme (the default). Unit conversions are performed automatically by the interface of the receiving community code.

needed, unit conversion of the values transferred between the models is automatically performed by
the interface of the receiving code, as explained in section 2.3.

Support for remapping between unstructured grids, is limited in the `CDO` library. Conservative interpolation of fields represented on unstructured mesh discretisations (Farrell et al., 2009) is being generalised in the `libsupermesh` library (libSupermesh, 2016) and could be utilised in the future.

### 2.5 State model

The internal work flows of different codes are in general not the same, even if they represent similar physics. This can be due to the differences in the algorithms or simply because of design choices. For example, a change in one of the grid variables may necessitate a reinitialization of variables in one code, while in another code this may not be needed. It is easy to add the corresponding functions for such reinitialization to the interface. The problem with this is that it introduces differences between
the interfaces, and is obviously error prone if controlled by the user. In order to manage this, the interfaces in `OMUSE` can be supplied with a representation of the work flow of a code. This is done in the form of a graph consisting of model states as the vertices and the transitions between them as the edges. Model states each have a set of allowable interface function calls. Such an interface call can trigger a transition between states (and for each transition there is a respective interface
function). With this *state model* `OMUSE` keeps track of the state of a, changing the state when needed (and calling the corresponding interface methods). The state model will change state automatically if an operation is requested that is not allowed in the current state. If the request can not be fulfilled an error is returned. The state model is flexible: states can be added and removed as required. Most codes can be made to conform to a simple state model similar to the six state model shown in Fig. 7.

### 2.6 Object-oriented interfaces

The object-oriented, or high-level, interfaces are the recommended way of interacting with the community codes. They consist of the low-level MPI interface to a code, with the unit handling, data model and state model on top of this. At this level the interactions with the code are uniform across




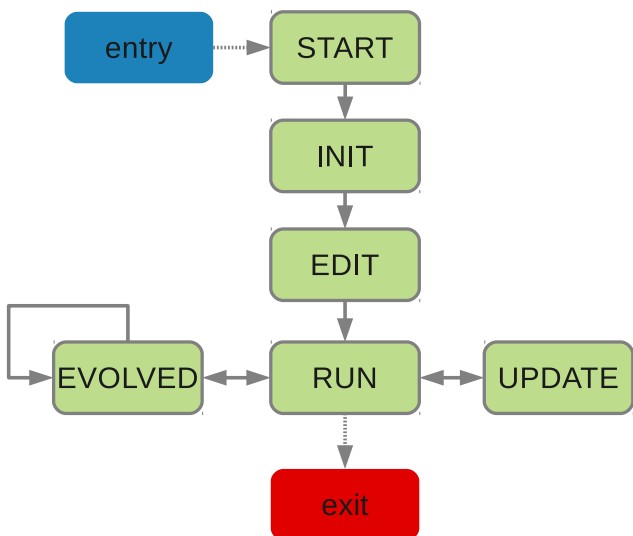

**Figure 7.** Example of a state model in OMUSE. The diagram gives the states that a simulation code can be in. Transitions between these can be triggered by explicit calls to the corresponding function (e.g. `initialize_code` from START to INIT) or implicitly (e.g. querying the grid state of a code may only be possible in the RUN state, and in this case the framework will call the necessary functions in order to get to the RUN state, guaranteeing a consistent state of the simulation code in the process). Adapted from Pelupessy et al. (2013).

different codes and the details of the code are hidden as much as possible. A lot of the bookkeeping
(arrays / unit conversion) is absent in the high-level interface formulation. This makes the high-level
interface much easier to work with and less prone to errors: the user does not need to know what
internal units the code is using, and does not need to remember the calling sequence nor the specific
order of calls.

### 2.7 IO

Community codes that are included into OMUSE will usually contain subroutines to read in and write
simulation data. This functionality is preferably not used within OMUSE. Instead, all simulation data
is to be written and read from within the OMUSE script (although in practice there can be reasons to
retain some of the original functionality as part of the interface). OMUSE includes a default output
format based on HDF5[3] that writes out all data pertaining to a data set, effectively standardizing the
IO for all the codes included in the framework. In order to simplify import and export of data, OMUSE
contains a framework for generic I/O to and from different file formats. A number of common file

---

[3]http://www.hdfgroup.org



formats used in the oceanographic and climate modelling community are implemented (ADCIRC grid files, netCDF), as well as generic table format file readers.

### 2.8 Data analysis

After a simulation, the generated data needs to be analyzed. Python has good numerical and plotting libraries available, such as Numpy and Matplotlib (Dubois et al., 1996; Hunter, 2007), and thus data analysis can be easily incorporated into the `OMUSE` workflow. While the simulation codes are running their internal state (as exposed through the interface) is accessible. This provides opportunities for efficient online data analysis, and also monitoring (or visualizing) the state of a running simula-

tion. Based on the state of the model, the simulations can also be scripted beyond what is originally implemented in the simulation code (examples of the latter are event-driven data output, or repeat simulation / resampling according to predefined conditions).

## 3 Component modules

In the present version, `OMUSE` contains an initial set of ocean models, namely `QG`, `ADCIRC`, `POP`

and `SWAN` (ideally one would like te reach a 'Noah's arc' milestone, Portegies Zwart et al. (2009), of having at least two independent application codes per domain). The implementation in `OMUSE` of the code interfaces is described in this section. The models cover different physics and / or a ranges of validity. and allow for are a number of different couplings between them. They also represent different levels of complexity in terms of code implementation, numerical schemes and a variety of

discretizations (described below). In addition to the simulation codes, `OMUSE` also contains support codes, including for example the `CDO` package introduced above in section 2.4.2 which is used to implement remapping schemes between different grids.

### 3.1 Simulation codes

#### 3.1.1 QG

`OMUSE` includes `QG`, a code to calculate the dynamics of quasi-geostrophic ocean flow. The flow on a $\beta-$plane with Coriolis parameter $f = f_0 + \beta_0 y$ is described by the barotropic stream function $\psi$ of the depth-integrated current velocity $\mathbf{u} = (u, v)$, with zonal velocity $u = -\partial\psi/\partial y$ and meridional velocity $v = \partial\psi/\partial x$. `QG` solves the governing barotropic vorticity equation (BVE) for $\psi$ (Pedlosky, 1996),

$$\frac{\partial}{\partial t}\nabla^2\psi + J(\psi, \nabla^2\psi) + \beta_0\frac{\partial\psi}{\partial x} = \frac{1}{\rho_0 H}\left(\frac{\partial\tau^y}{\partial x} - \frac{\partial\tau^x}{\partial y}\right) - R_H\nabla^2\psi + A_H\nabla^4\psi, \quad (1)$$

where the Jacobian $J$, here representing the advection of relative vorticity, is defined by

$$J(F, G) = \frac{\partial F}{\partial x}\frac{\partial G}{\partial y} - \frac{\partial F}{\partial y}\frac{\partial G}{\partial x}, \quad (2)$$



and $\tau = (\tau^x, \tau^y)$ represents the wind stress. `QG` can also solve for the first baroclinic mode of a mode expansion of the continuously stratified quasi-geostrophic vorticity equation (Flierl, 1978).

The parameters $\rho_0$ and $H$ are the reference ocean density and reference ocean depth, respectively. $R_H$ and $A_H$ are the bottom and lateral friction coefficients. `QG` solves (1) on a rectangular domain using a Cartesian grid. Boundary conditions consist of no-mass flux and/or no tangential stress (see for example Dijkstra and Katsman, 1997).

The `QG` code is written in Fortran 90 and uses the Poisson solver from the `fishpack`[4] or Intel

`MKL`[5] libraries (depending on compiler). Although conceptually simple, `QG` provides an instructive case study for importing a code in `OMUSE`, with its relatively simple internal state and without the complications of coordinate transformations, and serves as a template for other ocean models in `OMUSE`.

### 3.1.2   POP

The Parallel Ocean Program (`POP`) is a parallel global circulation model for ocean flows that solves the three-dimensional primitive equations for a stratified fluid using the hydrostatic and Boussinesq approximations (Smith et al., 2010). `POP` is often used to calculate strongly eddying ocean circulation models. However, resolving eddies on a scale that captures the instabilities that lead to ocean eddies requires the use of a high-resolution grid. Such high-resolution runs are computationally ex-

pensive, and `POP` is also frequently used for simulations at lower resolutions, in this case the effect of eddies is captured using sub-grid parameterizations (Gent and McWilliams, 1990).

The `POP` grid is a structured 2D grid in the horizontal dimensions, usually in a dipolar or tripolar configuration. `POP` requires that the grid dimensions are set at compile time. Therefore, we currently support two modes in which `POP` can be used through the `OMUSE` interface. The high-resolution

mode assumes a grid size of $3600 \times 2400$, corresponding to a $0.1°$ resolution. The low-resolution mode assumes grid dimensions of $320 \times 384$ horizontal grid points, corresponding to a $1.0°$ resolution with tropical stretching. Vertically, the grid contains 40 or 42 non-equidistant layers, increasing in thickness from several meters near the surface to 250 meters just above the lower boundary at 6000 meters.

`OMUSE` interfaces with a version of `POP` (based on version 2.1) that contains several extensions (van Werkhoven et al., 2014) [6]. This implementation includes a flexible load-balancing scheme and optionally uses Graphics Processing Units (GPUs) to accelerate compute-intensive parts of the code. Considering the fact that it takes at least 1000 simulated years to reach a near statistical equilibrium state, it is common practice to restart `POP` from a spun-up solution. The so-called 'restart file' and

other settings can be set through the `OMUSE` Python interface after the code has been instantiated and reached the 'START' state (see Fig. 7).

---

[4]`www2.cisl.ucar.edu/`
[5]`software.intel.com/en-us/intel-mkl`
[6]`https://github.com/NLeSC/eSalsa-POP`



As with all codes in `OMUSE`, the `POP` interface employs a state machine that tracks the model state and ensures consistency by automatically calling the appropriate transition functions in the low-level interface. To be able to set many of the configuration options through the Python interface

it was necessary to split several of the initialization routines in the `POP` source code. This was required because these routines used to read their configuration from a namelist file and immediately proceeded to initialize the model using that configuration. Within `OMUSE`, the model parameters are set through the interface as part of the Python script.

As such, the namelist file is only used to provide the code with default settings. After the settings

have been read from the namelist, the model halts and waits for the settings that are specific to the experiment to be passed through the interface. When the user has completed configuring the experiment, the state machine will automatically call a state transition function to complete the model initialization and advance the model to a state from which the user can interact with the model data or begin evolving the model.

The `POP` interface provides two different ways to supply the model with forcings, such as wind stress, surface heat flux, and surface freshwater flux. The first method is by setting the location of a file containing monthly averages of forcing data that will automatically be interpolated in time by the model. It is also possible to directly supply the model with forcing data through the interface, allowing `POP` to be coupled with, for example, an atmospheric model. When forcing data is supplied

through the interface, `POP` will not use data from file for that type of forcing.

In the `OMUSE` examples repository[7], we have included an example Python script for setting up a `POP` run in high-resolution mode in a cluster environment. The user script has to specify the location of the cluster head node and provide the requested number of nodes and cores and time required for the simulation. After that the user can instantiate the interface to create a running simulation and

interact with the model.

### 3.1.3 ADCIRC

The Advanced 3D Circulation model (`ADCIRC`) solves the shallow water primitive equations on a triangular unstructured mesh in either two or three dimensions. Water surface elevations $\zeta$, are obtained by solving the vertically-integrated continuity equation in the Generalized Wave Continuity

Equation (GWCE) formulation (Leuttich and Westerink, 2004). The momentum equations are either solved in vertically integrated form (2D mode), or in 3D (applying the Boussinesq and hydrostatic pressure approximations). In 3D, `ADCIRC` uses a generalized stretched vertical coordinate system (Leuttich and Westerink, 2004).

The `ADCIRC` mesh is represented in the `OMUSE` interface as an unstructured grid of nodes and

elements (which can be accessed as the `nodes` and `elements` attributes of an `ADCIRC` instance), representing the nodes and triangular elements of the grid. In the case of `ADCIRC` all prognostic

---

[7]`https://bitbucket.org/omuse/omuse-examples/`



variables (with the exception of the wet-dry status of elements) are defined by a linear $P_1$ finite element Galerkin representation over the entire domain, described by coefficients associated to mesh node positions. For example, in the simplest 2D case these are the water level, its time derivative

and the current velocities. The attributes of the elements are the nodes of each triangle, and its status (indicating whether an element is dry or wet). In addition to this, the interface defines a `forcings` grid, which accepts the (possibly time-dependent) forcings. Depending on the parameters of the simulation these can be for example wind stresses, atmospheric pressure, tidal potential, wave stresses etc. Boundaries are represented as sets of grids (one for each segment defined) with a reference to

the nodes in the boundary segment, a type attribute (describing the type of boundary) and any extra attributes necessary to specify the boundary condition (e.g. the water level for a boundary with prescribed elevations).

### 3.1.4   SWAN

In addition to the above models of hydrodynamical ocean circulation, `OMUSE` includes an interface
to `SWAN` (Simulating WAves Nearshore), a code to calculate the propagation of wind-driven surface waves (Zijlema, 2010, and references therein). `SWAN` uses a statistical description of the space and time varying wave properties, solving for the evolution of the action density $N(\boldsymbol{x}, t; \sigma, \theta)$, defined in terms of the wave energy density spectrum $E$ as $N = E/\sigma$, where $N$ is a function of space $\boldsymbol{x}$, time $t$, relative radian frequency $\sigma$ and direction $\theta$. The evolution of the action density is governed by the
action balance equation (e.g. Komen et al., 1994),

$$\frac{\partial N}{\partial t} + \nabla_{\boldsymbol{x}} \cdot [(\boldsymbol{c_g} + \boldsymbol{U})N] + \frac{\partial(c_\sigma N)}{\partial \sigma} + \frac{(\partial c_\theta N)}{\partial \theta} = \frac{S_{\text{tot}}}{\sigma}, \tag{3}$$

with $c_{\boldsymbol{g}}$ the wave group velocity, $\boldsymbol{U}$ the (depth averaged) current velocity, $c_\sigma$ and $c_\theta$ the propagation velocities in spectral and directional space, respectively. The source/sink term $S_{\text{tot}}$ represents the physical processes which generate, dissipate or redistribute wave energy. Amongst them, `SWAN`
includes generation of waves by wind, non-linear transfer of wave energy (including three- and four-wave interactions) and wave decay due to whitecapping, bottom friction and wave breaking (see SWAN, 2015, for more information).

`SWAN` discretizes (3) on rectilinear, curvilinear (structured) or unstructured (triangular) grids in one or two dimensions. The `OMUSE` interface to `SWAN` supports rectilinear and unstructured grids
(curvi-linear grids can be added). The type of grid, as well as the type of grid for the forcings are determined when the code is instantiated. Depending on the selected grid the interface defines a regular grid `grid` or an unstructured grid with `nodes` and `elements` attributes. These have an attribute to access the action density $N$ of the grid. In addition to this, the bathymetry can be specified and a number of potentially time-varying forcing inputs, like water levels, water current velocities
and wind velocities can be used (again a separate grid is used for the forcings).





To simplify the interface a few restrictions are placed on the forcings. For example, all the forcings in the interface use the same grid (whereas `SWAN` supports different grids for different forcings). This is not a limitation: within `OMUSE`, any regridding (if necessary because the sources of the forcings use different grids) can be done on the framework level. If both calculation grid and input grid are

unstructured, they are both assumed to use the same grid. In case of stationary calculations, the interface still defines an `evolve_model`, but it simply calculates the stationary action density (for all input times). It can still make sense to evaluate this in a time dependent fashion, as the input forcings (and thus the equilibrium state) may change with time.

### 3.2   Support modules

In addition to the simulation codes, support modules written in different languages can be included in `OMUSE`. Such a support module may, for example, provide functionality for coupling models. A support module can be interfaced with the same remote function interface as used for simulation codes. Currently, the only support module specific to `OMUSE` is `CDO` which is used for computing grid remapping weights and performing the remapping of quantities between different grids.

**3.2.1   CDO**

Climate Data Operators (CDO, 2015) is a command-line tool developed and maintained by the Max Planck Institute Hamburg containing over 400 operators that can process and manipulate climate data stored in self-describing file formats, such as netCDF.

    An `OMUSE` interface to `CDO` was created to be able to access the grid remapping functionality

within `CDO`. This library contains a reimplementation of the SCRIP package (Jones, 1999). The remapping weights computed by SCRIP are used by other climate model couplers, such as the Model Coupling Toolkit (Jacob et al., 2005), and OASIS (Valcke, 2013). In particular, the second-order conservative remapping scheme implemented in SCRIP is used to compute remapping weights for conservative exchanges of (e.g. heat and water) fluxes at the ocean-atmosphere interface.

A number of minor code modifications were necessary to be able to access the functionality in `CDO` as a library rather than as a command line tool. The low-level interface in `OMUSE` has to ensure that the internal state of `CDO` is consistent even though the code is not running as a command line tool. To do this, all grid information has to be propagated correctly to the different grid data storage structures used internally by `CDO`. In addition, the interface mimics some of the behavior of `CDO` to

produce the exact same results as when invoked from the command line. These include ignoring any land masks in the source and target grids and increasing the number of search bins in the computation of remapping weights.

    `OMUSE` implements a high-level object-oriented interface (called `CDORemapper`) on top of the low-level interface to `CDO`. This remapper can be initialized in three ways: (1) using a precomputed

weights file as produced by `CDO` from the command line, containing all information about the source





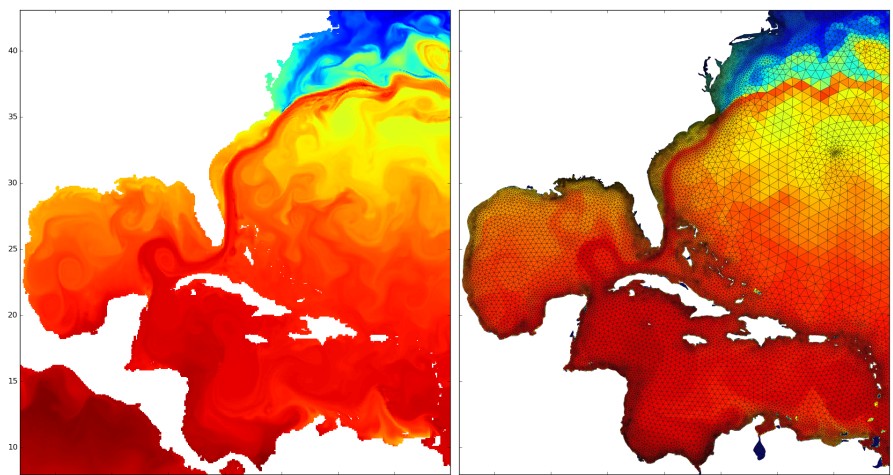

**Figure 8.** Result of a remapping performed by the `CDO` remapper using the `OMUSE` interface. A sea surface temperature field is remapped from `POP` using a $0.1°$ tripole grid (on the left) to the elements of an unstructured grid (on the right).

and destination grids, as well as the remapping weights, (2) using netCDF files for storing source and destination grid information (as used by `CDO` and **SCRIP**) and (3) setting `OMUSE` grid data types as source or destination grid. Modes (2) and (3) can be combined (if desired), and for these modes the remapping weights are computed automatically as the remapper initializes.

When using the default second-order conservative remapping scheme, the implementation of `CDO` also computes the gradients of the source field each time a quantity is being remapped. Note that the second-order conservative remapping scheme comes with limitations: the source grid has to be a structured grid because of the way SCRIP computes area integrals (for more information see the `CDO` documentation).

In figure 8 we show the result of a remapping performed by the `CDO` remapper using the `OMUSE` interface. A sea surface temperature field is remapped from `POP` using a $0.1°$ tripole grid to an unstructured grid. The second-order conservative remapping scheme was used to compute the remapping weights based on the grid information presented by the `OMUSE` interfaces of both simulations.

## 4    Code couplings

In addition to providing a unified interface to various types of codes, `OMUSE` has the objective of facilitating multi-physics simulations. For example, one would like to be able to couple a large-scale ocean circulation code with a regional ocean model (coupling across different scales), or couple a wave propagation model to an ocean flow model (coupling of different physics). Within `OMUSE`,





community codes can be combined into coupled models which have wider applicability than the
original codes. The setup of OMUSE allows for this in a transparent manner, such that the coupled
models have a similar interface as the individual models.

The types of coupling that OMUSE can be applied to is large, and range from simple input - output coupling to dynamic one-way coupling and to the development of two-way coupled solvers
(see more examples Pelupessy et al. (2013)). OMUSE provides the following features to facilitate the
building of coupled models: simplified, uniform access to the code simulation state, unified interfaces to the state of the simulation domain and its boundary conditions, and extensive automation of
bookkeeping operations.

### 4.1 QG model coupling

Some care is needed in the design of the code interfaces to ensure that couplings are as simple as pos-
sible. For example, the internal state of the QG simulation consists of the stream function $\psi$ on two
time levels, these are represented as a grid object with attributes psi, dpsi_dt and positions x and
y. It is more convenient to represent the two time levels as the (backward) time derivative dpsi_dt,
because this representation is independent of the time step (which can be different between codes).
The stream function $\psi$ (and its derivative) can also be queried at any position using an interface func-
tion get_psi_state_at_point . This function performs an (averaging) sampling and provides
a grid independent way to query and communicate the physical state. Another way to achieve this
would be to perform a copy using a remapping channel as described in section 2.4.2.

In addition, QG has two mechanisms to receive input from other codes: it calculates the evolution of the stream function using an input wind stress field. This wind stress field can be set by
changing the wind stress attributes tau_x and tau_y on the forcings grid. These can be copied
or remapped from another grid (read in from disk or generated dynamically by another code) or
by defining a (time and or position depend) functional form (from an analytic wind model, for example). Other possible inputs are the boundary conditions: $\psi$ and $\partial\psi/\partial t$ on the domain boundary.
These consist of four grid objects (one for each cardinal direction) of size $N_o \times 2$, where $N_o$ is the
number of grid points (in the corresponding dimension). Using these boundary grids it is possible
to implement two different strategies to vary the resolution over and/or the shape of the domain,
namely grid nesting and domain decomposition.

#### 4.1.1 Nested grid refinement

Depending on the parameters, equation (1) allows solutions with very narrow western boundary
currents. Numerically this presents a challenge as the required resolution at this boundary may be
much higher than for the rest of the basin. This is a typical situation where a nested solver (e.g.
Debreu and Blayo, 2008) may efficiently be employed. We can implement such a multi-grid coupled
solver within OMUSE using the base QG as an underlying engine. The solution of (1) is obtained on

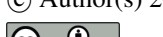



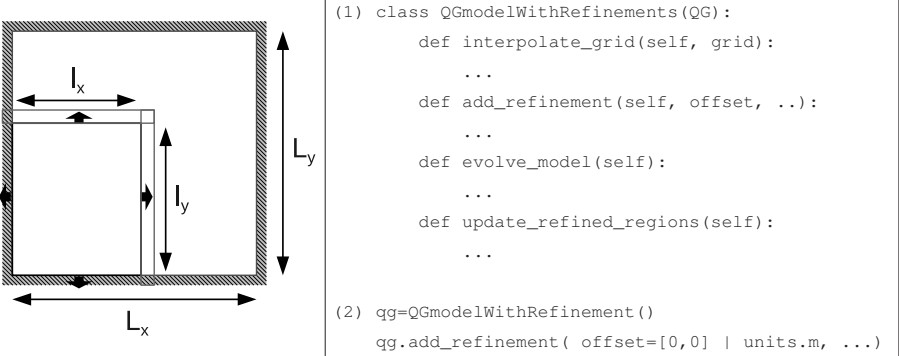

**Figure 9.** Schematic (left) and (abbreviated) definition of the refined `QG` model class (right) with an example (2) of its instantiation.

a base grid with a refined region of higher resolution where the two grids are solved by separate
instances of the `QG`.

Practically speaking, the following refinement strategy is followed (Fig. 9). Given a parent domain $L_x \times L_y$ a refined sub domain is defined by its offset, extension $l_x \times l_y$ and resolution dx. The low resolution region consists of the whole domain $L_x \times L_y$ (including the refined region). The `QG` is used to solve for the flow on $L_x \times L_y$. A second instance of the `QG` is used to solve the flow equation (1)
on the high resolution subdomain $l_x \times l_y$ given appropriate boundary conditions. This high resolution solution is then resampled and copied back (restriction operation) to correct the corresponding part of the domain on the low resolution grid.

If the boundary of the high resolution domain coincides with the boundaries of the parent domain (e.g. the east and south boundaries in Fig. 9) the boundary conditions are inherited from its parent.
Otherwise, the boundary of the high resolution region lies in the interior of $L_x \times L_y$, in this case $\psi$ and $\partial\psi/\partial t$ of the boundary can be obtained by interpolation of the low resolution grid. In our template implementation of this multigrid solver, we implement it as a derived interface in `OMUSE` (Fig. 9). It implements the same high-level interface (i.e. it has the same methods) as the base `QG`, which allows these two to be used interchangeably. In particular, a refined region can itself have
refinements.

### 4.1.2 Domain decomposition

Instead of overlapping domains, we can implement a similar coupling for (two or more) non-overlapping domains. A problem here is that the information used for the interpolated state on either side of a domain boundary does not carry information of the other domain. In the nested case the
low resolution solution is available over the whole domain, so it can provide this information.





This can be solved by iteration, but as the required step at each iteration (solving for $\partial\psi/\partial t$ using a Poisson solver) is quite expensive, this would be prohibitively inefficient. For this case, the problem can be accelerated by using accelerated vector extrapolation methods such as minimum polynomial extrapolation (MPE, Cabay and Jackson, 1976), i.e. we are solving for the fixed points of

$$\mathbf{x}^{k+1} = \mathbf{F}(\mathbf{x}^k), \tag{4}$$

where $\mathbf{x}^k$ is the vector consisting of the $\partial\psi_i/\partial t$ values on the boundaries (of all mutually neighbouring domains). In (4), $\mathbf{F}$ is the operator determining the next vector in this sequence, with iteration index $k$. This operator is provided by the instances of the QG, which calculates a new set of $\partial\psi/\partial t$ values from previous set. The MPE method does not need explicit knowledge of the sequence generator, and as such is especially well-suited for the problem here (this information in our case is 'hidden' in the QG code). In practice the solution converges within a handful of iterations to satisfactory precision.

The evolve loop of a compound QG consisting of N domains then proceeds as follows: (1) update the internal boundaries of each domain N. $\psi$ values are interpolated from neighbouring grids, a consistent set of $\partial\psi/\partial t$ values are calculated using the MPE method. (2) all the domains are stepped forward in time. An example of this will be shown in section 5.2 below.

Note that both preceding examples implement fairly close couplings. Nevertheless, the OMUSE framework can be used to implement these efficiently (both from the viewpoint of effort required to implement them as from a computational viewpoint. The most CPU intensive parts of the computations (i.e. the solutions to the BVE (1)) are executed by the (optimized) QG solver, while on the framework level a limited amount of bookkeeping operations and data transfer is handled.

## 5 Applications

To demonstrate the capabilities of OMUSE we present a number of example applications. These illustrate the application of the unified interfaces of OMUSE to calculate the same problem using different codes (section 5.1), the use of OMUSE to implement intra-code domain decomposition (section 5.2), a two-way coupling between codes with different physics (section 5.3), the embedding of a high resolution region in a low resolution domain using different codes (section 5.4) and the addition of data analysis to a running computation (section 5.5).

### 5.1 Critical transitions in a single-gyre ocean circulation model

The idealized classical model of a homogeneous mid-latitude wind-driven ocean (Sverdrup, 1947; Stommel, 1948; Munk, 1950) has been extensively studied using dynamical systems theory (e.g. Ierley and Sheremet, 1995; Sheremet et al., 1997), where the successive bifurcations in single-layer (constant density) models are analyzed as the parameters of the model are varied. Here we will use





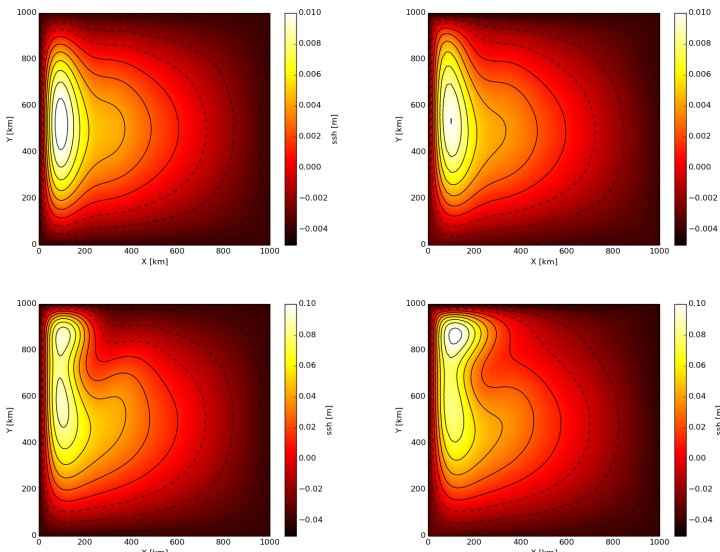

**Figure 10.** Comparison of `QG` and `ADCIRC` for a simplified mid-latitude ocean configuration. Shown is the equilibrium SSH for a square domain basin of equal depth, driven by surface wind stress using the setup of Viebahn and Dijkstra (2014) (resulting in a single gyre solution) at two different Reynolds numbers: $R = 1$ (top panels) and $R = 10$ (bottom panels), where $R = UL/A_H$ and $U = \tau_0/(\rho\beta_0 LH)$ is a characteristic horizontal velocity. In each case, the left panel shows the solution obtained using `QG`, and the right panel the `ADCIRC` solution is shown.

two completely different simulation codes to obtain equilibrium solutions and study the bifurcation
diagram in a single-gyre setup (Viebahn and Dijkstra, 2014).

The first code `QG` solves the BVE (1), while `ADCIRC` solves the primitive equations and does
not impose the quasi-geostrophic approximation. In this sense this simple numerical experiment
will illustrate a-posteriori the validity of the approximations made in deriving (1). We run the `QG`
simulation for a 1000 km×1000 km basin with a resolution of $N_o = 200 \times 200$ with parameters
$\beta_0 = 1.8616 \times 10^{-11}(\text{ms})^{-1}$ $R_H = 0 \,\text{s}^{-1}$, $A_H = 1194 \,\text{m}^2\text{s}^{-1}$, and a wind stress

$$\tau^x = -\frac{\tau_0}{\pi}\cos(\pi y/L) \; ; \; \tau^y = 0, \tag{5}$$

where $\tau_0$ is determined by the adopted Reynolds number $R = \tau_0/(\rho_0\beta_0 A_H H)$ ($\rho_0 = 1025 \,\text{kg/m}^3$
and $H = 4000 \,\text{m}$) For `ADCIRC`, a triangular grid matching this geometry is generated by subdividing
the cells of a ($N_o = 50 \times 50$) Cartesian grid into four triangles by adding a vertex to the center of
the cell. The parameters of `ADCIRC` are chosen to match the parameters in `QG`, and the same wind
stress is applied.



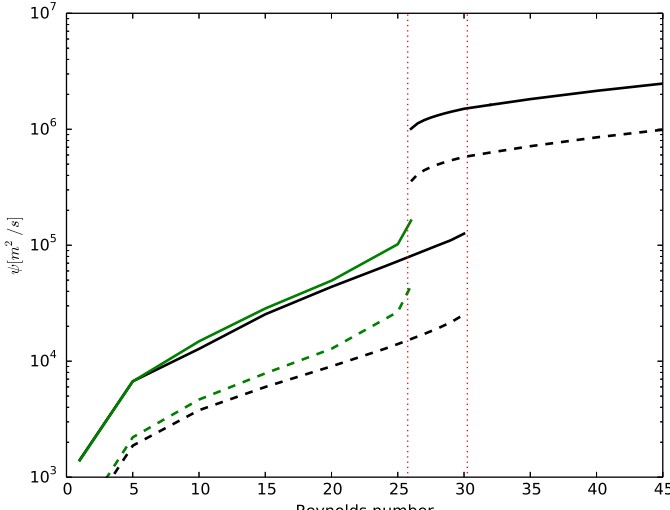

**Figure 11.** Part of the bifurcation diagram showing the upper and lower branches of steady and oscillatory solutions for a single gyre ocean model. Shown are the mean (dashed) and maximum (solid) value of the stream function for `QG` (black) and `ADCIRC` (green) model runs, as a function of the Reynolds number $R$. For `ADCIRC` the stream function is calculated as $\psi = g\zeta/f_0$, where $\zeta$ is the free-surface height. The values shown represent time averaged values in case the system shows oscillatory behaviour. The flow undergoes a cyclic fold bifurcation near $R = 25$ as indicated by the vertical dashed lines (Viebahn and Dijkstra, 2014). The `ADCIRC` solution becomes (numerically) unstable at this bifurcation.

In Figure 10 we compare the stable stationary solutions of the two codes (these are obtained by running until the maximum fractional changes in either stream function $\psi$ (for `QG`) or sea surface elevation $\eta$ (for `ADCIRC`) between two successive diagnostic time intervals changes less than $10^{-4}$).

As can be seen, the two codes calculate solutions that agree well (although small differences can be seen). Figure 11 shows the corresponding bifurcation diagram when varying the Reynolds number. The correspondence between the two codes is good for low Reynolds number, showing the same qualitative behaviour. At the bifurcation (above $R \approx 25$) we found that the solutions obtained by `ADCIRC` become unstable to a basin-wide fast gravity wave mode, which is not represented in the

`QG` model.

## 5.2   `QG` on a composite domain

As a first example of the use of `OMUSE` to construct new solvers by composing various subcodes, we show the results of an idealized calculation solving the BVE (eq. 1) on composite domains. The





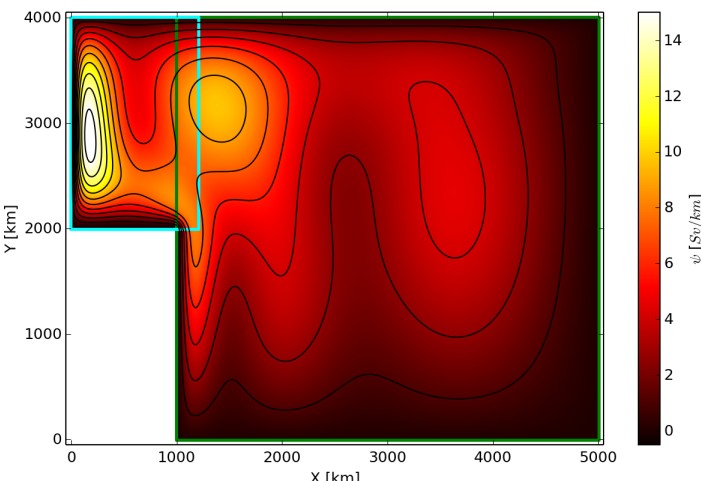

**Figure 12.** Stream function $\psi$ for a non-rectangular domain run with `QG` on a composite domain. Plotted is $\psi$ after 15 days of evolution with the composite `QG` code (section 4.1.2) on a domain consisting of two coupled subdomains, indicated by the cyan and green rectangles.

coupled solver presented in section 4.1.2 is employed for this. It uses separate instances of `QG` to

calculate the ocean flow (i.e. solutions to equation (1)) for a composite domain. In figure 12 the solution is calculated on a domain with a western boundary that is stepped. The domain (shown in Figure 12) consists of a 4000×4000 km basin extended on the western side with a 1200 ×2000 km subdomain (the respective submains are indicated in the figure by the green and cyan rectangles). The solution is shown for a Reynolds number $R = 10$, with similar single gyre forcing as (5) after

15 days of evolution (at this early stage one can distinguish the Rossby waves moving east to west from the interior of the large basin, into the smaller domain).

Using such a composite domain it is possible to calculate the effects of topographic features on the dynamics of boundary currents, or change the resolution across the domain. Such idealized modelling on a simplified domain is often useful to reduce the real world topography to its essential fea-

tures, e.g. Le Bars et al. (2012). The example above implements a tailored solver using the high-level `OMUSE` interface to `QG`. This demonstrates that the interfaces of `OMUSE` are capable of expressing fairly tight couplings. The alternative, and maybe more obvious, way to implement such solver is to adapt the underlying Poisson solver to various domain shapes, which may involve changing the data representation. In contrast, the implementation here is done without reference to the underlying data

structures and in principle does not depend on the grid type or shape used in the underlying solver.





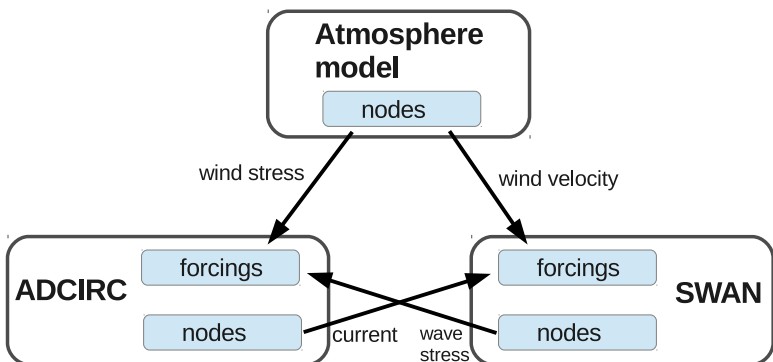

**Figure 13.** Schematic representation of the `ADCIRC-SWAN` coupling.

```
(1) channel1=hurricane.grid.new_channel_to( swan.forcings )
(2) channel2=hurricane.grid.new_channel_to( adcirc.forcings )
(3) channel3=adcirc.nodes.new_channel_to( swan.forcings )
(4) channel4=swan.nodes.new_channel_to( adcirc.forcings )
(5) while time<tend:
(3)     hurricane.evolve_model(time+dt/2)
(4)     channel1.copy_attributes(["tau_x","tau_y"])
(5)     channel2.copy_attributes(["vx","vy"])
(6)     adcirc.evolve_model(time+dt/2)
(7)     swan.evolve_model(time+dt/2)
(8)     channel3.copy_attributes(["current_vx","current_vy"])
(9)     channel4.copy_attributes(["wave_tau_x","wave_tau_y"])
```

**Figure 14.** Definition of communication channels and evolve step corresponding to figure 13.

### 5.3 Implementation of a coupled SWAN-ADCIRC model

The propagation of wind-driven surface waves is sensitive to water levels and current velocities. The properties of the underlying circulation will affect the evolution of the wind-driven wave field and the location of wave-breaking zones. On the other hand, wind-driven wave transport can generate

radiation stress gradients that can in turn drive circulation set-up and currents. Currents can also be affected by changes in the vertical momentum mixing and bottom friction stresses generated by the wind-driven wave field. Thus, in many coastal applications, such as the calculation of storm surges, waves and circulation processes should be mutually coupled.

Here we will demonstrate the implementation of such a coupling within the `OMUSE` framework,

applying it to a coupling of the `ADCIRC` circulation model and the `SWAN` wave propagation model. A fully integrated coupled `ADCIRC`/`SWAN` model exists (Dietrich et al., 2011), and below we compare and contrast our method of coupling with this existing approach. The physical interactions between the different simulated components are schematically given in Fig. 13. Figure 14 shows the (some-




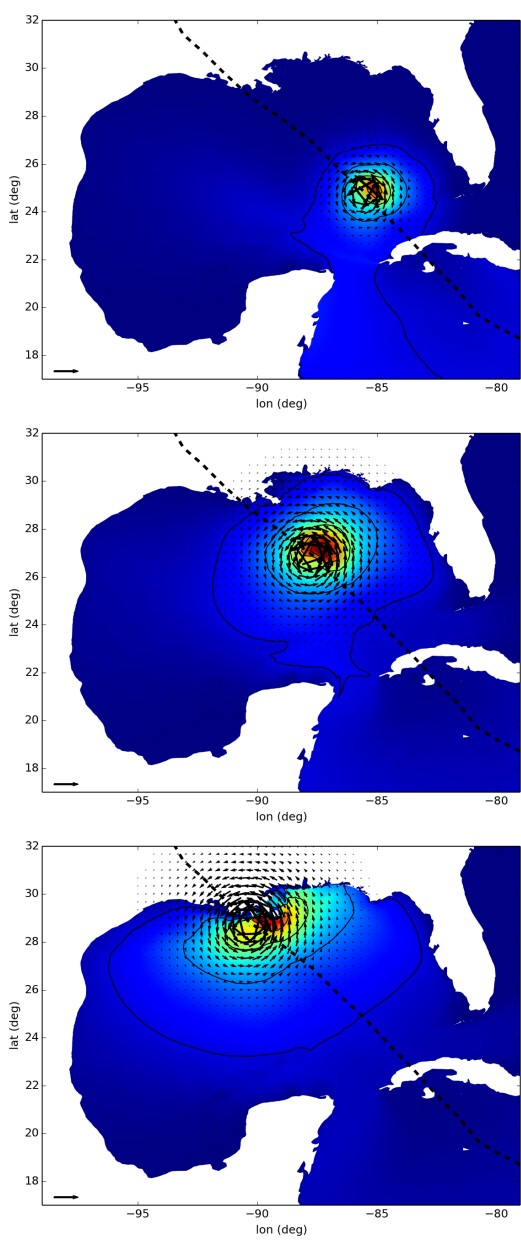

**Figure 15.** Significant wave heights for hurricane Gustav (2008), calculated using a coupled `ADCIRC-SWAN` simulation. The significant wave height field (shading, with contours at 1, 3, 6, 9 and 12 meters) is shown with the (model) wind field superimposed (arrows, where the arrow on the lower left corresponds to 30 m/s), and the storm track (dashed line). Shown are frames 156, 168 and 180 hours after start of the simulation (2008/08/25/0000 UTC), in the three panels from top to bottom, respectively.





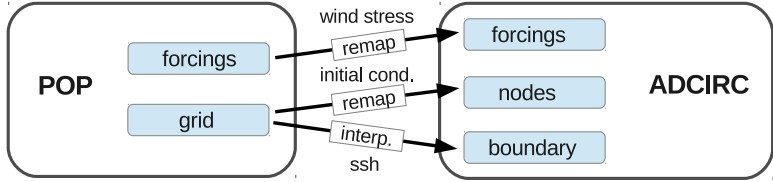

**Figure 16.** Schematic representation of the `POP-ADCIRC` one way coupling for an embedded domain. The labelled arrows indicate the use of remapping channels. "remap" stands for a conservative remapping between the structured `POP` grid and the unstructured `ADCIRC` grid, while "interp." indicates that the variables are interpolated.

what simplified) `OMUSE` code corresponding to this model coupling. Note that in this coupling both
`SWAN` and `ADCIRC` use the same unstructured (triangular) grid. The communication between the codes (as shown in Fig. 14) is handled by `channels`, whereby the framework handles the copying (and unit conversion) of data.

As an example we apply the coupled code to calculate the wave height and storm surge of hurricane Gustav (2008) [8] in the Gulf of Mexico. The hurricane is modelled using an analytic prescription
(Holland, 1980) from data of a hurricane storm track (positions, central pressures, maximum windspeed, storm radius) read in from file. Implementation of this analytic model is in the form of a Python class mimicking a full simulation code. `ADCIRC` is run in 2D barotropic mode with meteorological forcing from the hurricane model and `SWAN` provides the wave stresses. There is no forcing on the open ocean boundaries. For the discretization of the action density, `SWAN` uses 36 bins in the
directional space and 32 bins in frequency (from 0.05 to 1 Hz). The standard set of third generation wave parameters, including the effects of wave breaking, bottom friction and 3-wave interaction is used. The time step (`dt`) between updates of the coupled quantities is 600 seconds.

In figure 15 we show the resulting wave heights calculated by the model during the development of hurricane Gustav at three different times. The results of the `OMUSE` coupling are similar to the re-
sults of the integrated coupling implementation (Dietrich et al., 2011, and above mentioned website). Technically the coupling as in `OMUSE` differs from the implementation by Dietrich et al. (2011), as the latter directly copies data in the unified memory space of a single binary (an for that reason is more efficient). However, both implement the same coupled processes and the approach taken by `OMUSE` does not depend on the particular aspects of the selected codes - exactly the same script
could be used by other codes using the same interfaces.





```
(1) forcings_channel=pop_forcings_grid.new_remapping_channel_to(
                    adcirc.forcings, conservative_spherical_remapper)
(2) boundary_channel=pop_grid.grid.new_remapping_channel_to(
                    adcirc.elevation_boundary, interpolating_remapper)
(3) while time<tend:
(4)     pop.evolve_model(time+dt/2)
(5)     forcings_channel.copy_attributes(["tau_x","tau_y"])
(5)     boundary_channel.copy_attributes(["ssh"])
(6)     adcirc.evolve_model(time+dt)
(7)     pop.evolve_model(time+dt)
(8)     time+=dt
```

**Figure 17.** Definition and use of remapping channels for the `POP-ADCIRC` embedding of figure 16.

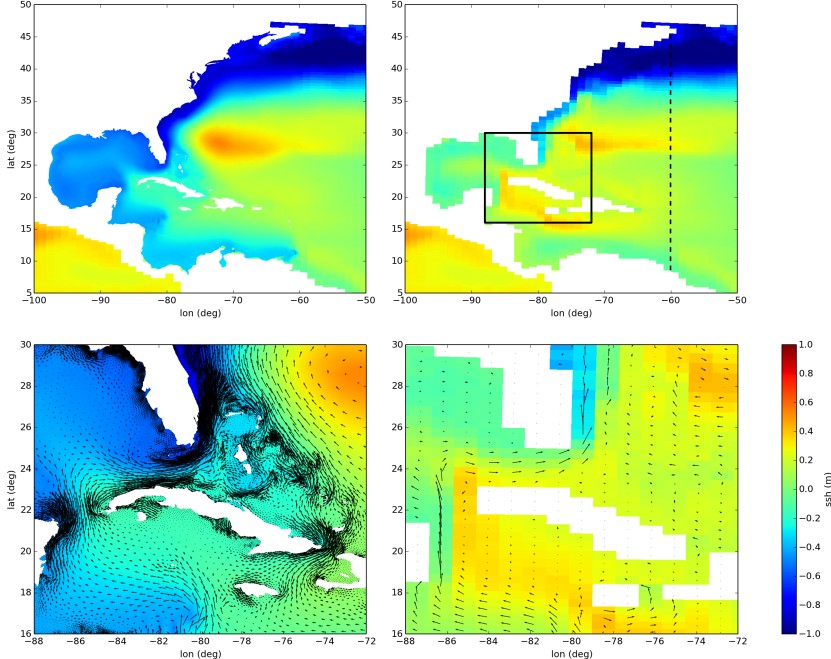

**Figure 18.** Sea surface heights and velocities of a `ADCIRC` run embedded in a global circulation `POP` model. Top panels show the sea surface height (SSH) of a region covering the Western North Atlantic Ocean, Caribbean Sea and Gulf of Mexico. The upper left panel shows the high resolution `ADCIRC` SSH field (superimposed on the `POP` field) and the upper right panel the low resolution `POP` field. The black square indicated in the top right panel is shown in more detail in the lower panels where the SSH with velocities superimposed are shown (in the case of `ADCIRC` the barotropic velocities are shown, for `POP` the are the surface velocities). The dashed line (top right panel) is the open ocean boundary of the regional `ADCIRC` model.





### 5.4 Embedded regional model

A recurring problem for regional or coastal modelling is the application of realistic boundary conditions from the open ocean, even more so when one is interested in the effect of large scale or global processes on the regional level. One approach to obtain realistic boundary conditions at the required scale is the nesting of a high resolution and small scale model in a lower resolution but larger scale model (e.g. Debreu et al., 2012; Djath et al., 2014).

Here we illustrate the implementation of (one-way) nesting in `OMUSE` by embedding a regional high resolution barotropic `ADCIRC` model of the Caribbean and North American Atlantic coast into a `POP` global circulation model (see fig. 16). In this case, since `POP` uses a curvilinear structured grid and `ADCIRC` an unstructured triangular mesh, it is necessary to perform a remapping when transporting variables from one code to the other (these functional *remapping* channels are indicated in figure 16 by the labelled arrows).

For the actual implementation of the coupling in `OMUSE`, the difference between using a remapping channel and a normal (data copying) channel (such as the ones used in section 5.3) is small: the only difference with a normal channel is that upon initialization the actual remapping method to be used needs to be specified for a new remapping channel. The usage of the remapping channel to prescribe the data flow in the coupled model (figure 17) uses the same semantics.

In order to calculate the dynamics of the nested regional model, `ADCIRC` in 2D barotropic mode needs an input wind stress field and the specification of either the sea surface level or normal fluxes on the boundary. In addition to this, the model can be initialized from remapped flow variables (barotropic velocities and sea surface heights). Note that a fully consistent coupling between the two codes is not possible since they solve for a different set of variables (2D barotropic vs 3D baroclinic). For the (conceptual) example here, a coupling was made on the sea surface elevation, and the bathymetry of the `ADCIRC` grid was limited to 500m depth (so the barotropic basin represented in `ADCIRC` can only be compared with the upper 500m layer of `POP`). The time step for the coupling (updates of the boundary surface elevations) is taken to be equal to the `POP` internal time step of approximately 30 minutes. The remappings are performed at each time step for the wind stresses and for the sea surface heights.

Figure 18 shows the sea surface heights and velocities on the original low resolution `POP` grid and the embedded higher resolution `ADCIRC` grid after 30 days of adjustment (after this the `ADCIRC` solution follows the (slow) variations of `POP`). A fully consistent coupling is possible when using `ADCIRC` in baroclinic mode. In this case, the coupling proceeds (with a larger number of coupling variables involved) along similar lines.

---

[8]The data for this example comes from:
http://www.caseydietrich.com/2012/06/27/example-input-files-for-swanadcirc/



### 5.5 On-the-fly data analysis

In addition to consuming massive amounts of CPU time, current large scale simulations are capable
of generating enormous amounts of data. Usually, it is possible to store only a very limited subset
of this data, this limits the data analysis that can be performed. One solution to this has been to do
(part of) the analysis on the fly. Online data analysis offers several opportunities, including the fact
that special actions can be taken when interesting events occur. Such special actions may include

inspecting the model internal data at resolutions, both spatial and temporal, that are not available or
feasible with offline data analysis. While running simulations through `OMUSE`, the simulation state
is accessible, and this allows for data analysis while a simulation is running.

As a proof-of-concept application we add an online ocean eddy tracker on top of the `POP` model.
The interest in ocean eddies comes from the fact that eddies transport considerable energy and

mass and as such influence the dynamics of large-scale ocean circulation and the climate (e.g.
Viebahn and Eden, 2010; Griffies et al., 2015). To understand eddy properties and variability, several
mesoscale eddy tracking algorithms have been proposed in recent years. We have adapted a sea sur-
face height-based eddy tracking code that is implemented in Python, called `py-eddy-tracker`
(Mason et al., 2014). The code uses high-pass filtered sea level anomaly (SLA) fields. On the filtered

fields, contours are computed at 1 cm intervals for levels between -100 cm to 100 cm. These contours
are then searched to locate eddies based on their shape, area, and amplitude. `py-eddy-tracker`
tracks eddies across successive sea level anomaly (SLA) fields using a search ellipse, bounded by
the local (long baroclinic) Rossby wave speed.

We have generalized the code in order to use different data sources, including output that is ob-

tained directly from numerical models. To this end, we have modified the `py-eddy-tracker` to
be able to handle grids that contain gaps, as land-only blocks are not part of the simulation in POP.
We use `Basemap`[9] to compute a landmask for the given grid and apply it to the SLA field. Finally,
we have created a simple, but easy to use, interface to the `py-eddy-tracker` that understands
the grid data structures and units used in `OMUSE`.

Figure 19 shows the code required to build an online eddy tracking program with `OMUSE`. The
interface `EddyTracker` is given the `OMUSE` grid datatype used by `POP` and automatically per-
forms unit conversions and extracts the information that it needs (i.e. the sea surface height and the
coordinates of the grid points).

Figure 20 shows the output of the online eddy tracking program that uses sea surface height

data directly from a running `POP` simulation. In this image, we can clearly see the large anticyclonic
eddies that result from the retroflection of the Agulhas Current, as well as many smaller eddies being
tracked over time by the online eddy tracking algorithm. The data generated by the online eddy
tracker can, for example, be used to compare the statistics of the simulated eddies to the analysis
made using `py-eddy-tracker` (or other tools) of altimetry data.

---

[9]`http://matplotlib.org/basemap/`





```
from omuse.ext.eddy_tracker.interface import EddyTracker
from omuse.community.pop.interface import POP
p=POP( ... ) #start POP as you would do normally

dt_analysis = 7 | units.day
tracker = EddyTracker(grid=p.nodes, domain='Regional',
    lonmin=0. | units.deg, lonmax=50. | units.deg,
    latmin=-45. | units.deg, latmax=-20. | units.deg, dt_analysis)

tnow = p.model_time
stop_time = p.model_time + (1 | units.yr)

while (tnow < stop_time):
    p.evolve_model(tnow + dt_analysis)
    tracker.find_eddies( ssh=p.nodes.ssh, rtime=p.model_time )
    tnow = p.model_time

tracker.stop(tend)
p.stop()
```

**Figure 19.** This example demonstrates how to build an application that analyzes data from a running simulation using `OMUSE`. This code implements an online eddy tracking program that tracks the eddies based on sea surface height every seven days for one year of `POP` simulation.

**6 Summary and Discussion**

We have presented the Oceanographic Multipurpose Software Environment (`OMUSE`) which provides a homogeneous interface to existing or newly-developed ocean models. As illustrated by the results in the previous section, the use cases for `OMUSE` range from running simple numerical experiments with single codes (e.g. section 5.1), to combining simulation codes and data analysis tools

(section 5.5) and setting up fairly complicated and strongly coupled solvers (section 5.2) to solve problems that are intrinsically multi-scale (section 5.4) and/or require different physics (section 5.3). Using `OMUSE`, simulations can be easily scripted and on-the-fly data-analysis can be added.

The implementation of the different use cases is facilitated by several aspects of the `OMUSE` design. `OMUSE` defines standardized interfaces and data structures for different codes. The data struc-

tures and the state model as well as the communication model used in `OMUSE` are flexible and allow a wide variety of codes, written in different languages, to be integrated with `OMUSE`. `OMUSE` also works well with established methods to generate initial conditions and analyze the resulting data.

`OMUSE` shares some of the goals of a number of other coupling frameworks that have been developed in the earth system modelling community (e.g. Hill et al., 2004; Buis et al., 2006; Gregersen et al.,

2007; Jacob et al., 2005; Larson, 2005; Peckham et al., 2013; Valcke, 2013). The closest equivalent is the Community Surface Dynamics Modeling System (`CSDMS`; Peckham et al., 2013). `CSDMS` and `OMUSE` follow a similar design philosophy (as summarized in Peckham et al. (2013)), by aiming for a modular component based modelling framework. This similarity translates, in principle, into





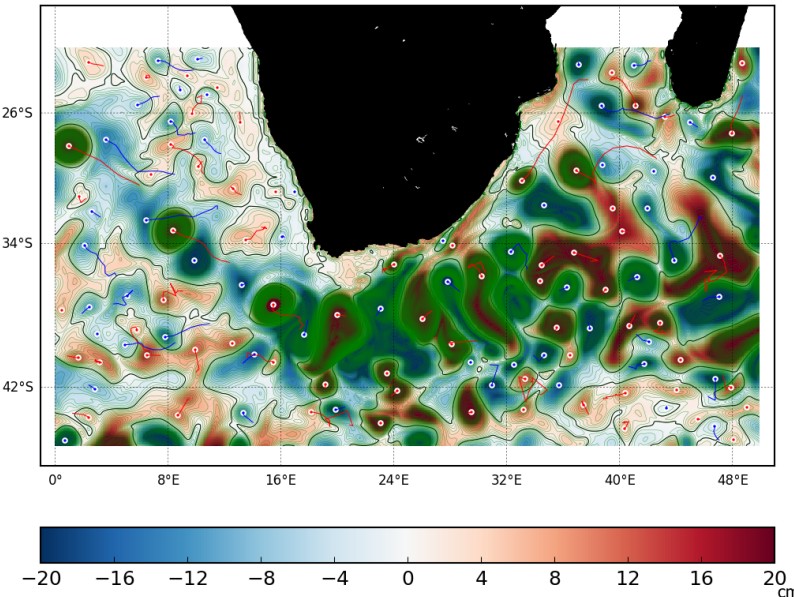

**Figure 20.** Output of the online eddy tracking application using data from a running `POP` simulation, showing a region around the southern tip of Africa. The green lines show the contours between areas of different sea level anomaly values. Red indicates areas of elevated sea level, and is used to detect anticyclonic eddies. Similarly, blue indicates a lower sea level, and is used to identify cyclonic eddies. The red or blue lines indicate the track that an eddy has travelled since it was first detected.

inter-operability since the interface components of the `CSDMS` could be easily adopted for an `OMUSE`
interface (and possibly vice versa). The `CSDMS` BMI (basic model interface) and CMI (component model interface) are roughly equivalent to the `OMUSE` low and high-level interfaces, respectively The main differences between `OMUSE` and `CSDMS` are that the former presents Python as the main user interface for programming an application, while for the `CSDMS` there are various choices, including a GUI frontend. In addition, `OMUSE` simplifies the interaction with the community codes
using high-level object-oriented data structures and `OMUSE` has a more extensive and flexible state model.

It is important to ensure the accuracy, reliability and reproducibility of a integrated framework like `OMUSE`. We employ a number of strategies to ensure this is the case. The framework itself is tested daily and upon the commit of changes using more than 2000 component tests that cover approxi-
mately 80% of the framework code and range from basic tests of the interfaces to the simulation codes as a whole. The simulation codes themselves are validated by comparing the results of test problems run using `OMUSE` with the results of the code running stand-alone (usually a number of test problems are developed for the simulation codes). In some cases (for example the `ADCIRC`-



SWAN coupling) the results of a coupled solver implemented within OMUSE can be compared with a
reference coupling implementation (Dietrich et al., 2011, e.g.). In any case, to ensure the correctness
of a new application in OMUSE one should conduct the usual tests to ensure the validity and verify
the results.

An important concern of a coupling framework such as OMUSE is performance. While the initial
driver for the development of OMUSE is to simplify the setup and development of coupled simu-
lations, the architecture of OMUSE is designed with a high degree of parallelism. The internal data
structures are efficient. Also the individual simulation codes are often highly optimized. So the per-
formance of an OMUSE application is rarely a concern, but this is strongly problem dependent. In
practice, the overhead imposed by the framework is often measured to be rather small (less than a
few percent), but it is not difficult to formulate problems where the strength of the coupling is intrin-
sically so strong that very frequent communication between the component solvers is necessary.

In this respect a limitation of the current design of OMUSE is the fact that the communication
between solvers is handled by the master script. This imposes a bottleneck for the performance
of the communication between e.g. two parallel codes. While in the current setup there are some
mitigating techniques that can be applied (asynchronous communication or grouping and spawning
the communication-intensive subprocesses), ultimately we would need to implement a *distributed*
communication channel that would direct the data flow from the sending to the receiving process
directly. Note that such distributed communication channels would not change the semantics of the
use of a channel between data structures.

**Code availability**

The main framework and community modules are production ready. OMUSE is foreseen to grow over
time with new codes and capabilities. OMUSE is freely downloadable [10] and comes with a testing
framework and basic examples. Furthermore, it can easily be adapted for private use (the licence is
GPL3).

We distribute the simulation codes that are interfaced by OMUSE together with the framework, if
the authors distribute their code with an open source licence, otherwise these codes must be down-
loaded separately. New codes or extensions, as well as bug fixes may be submitted to the repository.
OMUSE encourages the practice of distributing simulation codes by reporting automatically, upon
conclusion of an OMUSE script, which community codes were used during the run and suggesting
references for inclusion in any publications.

**Extending OMUSE**

The effort required to import or interface a code within OMUSE varies with the code complexity, and
depending on whether a similar code already exists within the framework (in this respect the codes

---

[10]https://bitbucket.org/omuse/omuse





already included provide a good starting point). In order to be interfaced, a code needs to be written in a programming language for which MPI or socket bindings are available. The complete procedure
(along with examples) is described in detail in the documentation section of the source distribution and the project website; here we only briefly outline the procedure.

To import a community code, one first creates a directory in the OMUSE community code base directory with the name of the module. The original source tree is imported in a subdirectory (by convention named 'src'). The top-level directory contains the Python side of the interface
('interface.py'), the interface in the native language of the code (e.g. 'interface.c') and a file for the build system ('Makefile').

The Python interface (described in the file interface.py) typically defines two classes, the low-level interface and the high-level interface. The former contains the function definitions of the calls which are redirected through the MPI communications channel to the corresponding call de-
fined in the native interface file (interface.c). The high-level interface defines the units of the arguments of the function calls (see section 2.3). In addition it specifies the parameters of the code, the state model (section 2.5) and the mapping of the object oriented data types to the corresponding low-level calls. By default, the data of the simulation is maintained in the community code's memory (and accessed transparently as described in section 2.4).

For modern and modular codes, often no or little changes in the original source code base (in 'src') are needed. In other cases, a code may need significant source code changes (e.g. to seperate the initilization stages and timestepping) or additions to implement functionality that is required for the OMUSE interface (e.g. externally imposed boundary conditions for grids). In these cases more effort is required to import the code and this will also make it more difficult to update the interface
to a new version of the community code.

In our experience writing an interface to a new code, which also involves writing tests, testing and debugging the interface, represents a modest amount of work. While every code is different and has its own peculiarities, it is typically something that can be completed (by someone with some familiarity with the source code) during a short working visit or small workshop. Defining an
interface for a new domain (exposing new physics) can take longer, as these need refinement over time.

*Acknowledgements.* OMUSE was developed as part of the ABC-MUSE project, funded by Netherlands eScience Center (file number 027.013.701, 2013-2016). This research was supported by the European Union's Horizon 2020 research and innovation programme under grant agreement No 671564 (COMPAT project) and by the
Nederlandse Organisatie voor Wetenschappelijk Onderzoek (Netherlands Organisation for Scientific Research, NWO) under project number 858.14.062. We want to thank Marcel Zijlema and Julie Pietrzak for discussing and commenting on the manuscript.



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
