# Peer review of "The Oceanographic Multipurpose Software Environment (OMUSE v1.0)"

_Geoscientific Model Development, 2016_

## Short Comment (SC1) · 13 Sep 2016

Dear authors,

In my role as Executive editor of GMD, I would like to bring to your attention our Editorial version 1.1:

http://www.geosci-model-dev.net/8/3487/2015/gmd-8-3487-2015.html

This highlights some requirements of papers published in GMD, which is also available on the GMD website in the 'Manuscript Types' section:

http://www.geoscientific-model-development.net/submission/manuscript_types.html

In particular, please note that for your paper, the following requirements have not been met in the Discussions paper:

[Figure]

- "The main paper must give the model name and version number (or other unique identifier) in the title."

- "If the model development relates to a single model then the model name and the version number must be included in the title of the paper. If the main intention of an article is to make a general (i.e. model independent) statement about the usefulness of a new development, but the usefulness is shown with the help of one specific model, the model name and version number must be stated in the title. The title could have a form such as, "Title outlining amazing generic advance: a case study with Model XXX (version Y)"."

In order to simplify reference to your developments, please add a version number and consider to add the models acronym in the title of your article in your revised submission to GMD.

Additionally, I like to point out, that the Code Availability and the Acknowledgement Sections are distinctive parts of the overall paper structure (see the section "manuscript composition" on http://www.geoscientific-model-development.net/for_authors/manuscript_preparation.html). In contrast to this, "Extending OMUSE" is not part of this overall structure. Therefore this section is in the wrong place. It should be either a section in the main body of the paper or an appendix section, but it should definitely not be placed between Code Availability section and Acknowledgements .

Yours,

Astrid Kerkweg

―――――――――――――――――――

---

## Author Comment (AC1) · 14 Sep 2016

Thanks for pointing this out, we will provide the version number and move the section "Extending OMUSE" at the next revision,

regards,

Inti Pelupessy

---

## Referee Comment (RC1) · Anonymous Referee #1 · 14 Oct 2016

General comment

This paper presents the Oceanographic Multipurpose Software Environment (OMUSE), its design, model and support components, and illustrates its use by presenting few diverse applications. OMUSE is a python scripting framework that allows running numerical experiments involving different, possible coupled, ocean models and different tools for on-line analysis or post-processing of these ocean model results, on possibly heterogeneous computing resources. This paper is clear and well written and the discussion part includes a comparison with other existing frameworks and some reflections about the current limitations of the system. As such, I consider it deserves publication after considering the minor revisions proposed here.

Specific comments

- Title: Please provide more details about OMUSE in the title (not just its name) so to better reflect the content of the paper. Also, it is recommended to provide in the title a version number of the latest OMUSE version available (see also A. Kerweg's comments).

- L15: I am not sure the application examples described show the efficiency of OMUSE; I would insist more on the flexibility but less on the efficiency.

- L63: You write "This has the benefit of the parallelism and . . ." . This seems incoherent to me with what you write in the discussion L716-L717: "limitation of the current design of OMUSE is the fact that the communication between solvers is handled by the master script. This imposes a bottleneck for the performance of the communication between e.g. two parallel codes." Also, L105: What is the benefit of providing built-in parallelism in the MPI-based remote function protocol if, as written in the discussion the handling of the communication through the master script imposes a bottleneck? Can you clarify?

- L64 and other places: You mention the "bookkeeping inherent to code coupling". Can you explain what you mean by bookkeeping? In particular, what do you mean precisely L441-442 by "extensive automation of bookkeeping operations"?

- L116: What do you mean by "the sockets channel is mainly useful for cases were a component process is to be run on one machine". Of course, a process is always run on a machine! Do you mean" "the sockets channel is mainly useful for cases where a component process is to be run a different machine".

- L126: you write: "the communication requirements between processes must not be too demanding. Where this is not the case (e.g. when a strong algorithmic coupling is necessary) a different approach may be more appropriate." Why should the (MPI) communication be "not too demanding"? What do you mean by "a different approach"? Do you mean something else that OMUSE?

- L190-191: I am not sure I understand the difference between Cartesian and Regular.

What do you mean by "constant"? constant in time (I suppose not)? Constant with respect to the dimension? Please clarify.

- L241: can you clarify what you mean by "This functionality is preferably not used within OMUSE." Does OMUSE support code using their own I/O library or not? The word "preferably" is ambiguous.

- L385-388: What is the relation between these two sentences (starting with "In case of stationary ...") and the restrictions on the forcings discussed in this paragraph?

- L401-402: SCRIP library is included in OASIS but it is not in MCT, even if MCT can use grid remapping weights and addresses generated (separately) by SCRIP.

- L422-423: the source grid has to be structured not because of the way SCRIP computes the area integrals but for the calculation of the gradients needed for the 2nd order.

- L486: Section 4.1.2 is about non-overlapping domains. Why is it named "Domain decomposition"? Also, are you talking here really about "non-overlapping" domains or more precisely "partially-overlapping" domains, as in the 5.2 example?

- L507: What do you mean by "preceding examples"? Are these the 4.1.1 and 4.1.2 couplings? If so, maybe put the numbers 4.1.1 and 4.1.2 for clarity.

Minor comments:

- L14: Remove the , after "solver"

- L23: I would not write "current CMIP5" as CMIP5 is over now and CMIP6 is on-going.

- L123: maybe replace "the requested subroutine calls" by "the requested simulation code subroutine calls" (if I got this right)?

- L144, change "OMUSE implements a" for "OMUSE implements also a"

- L509, add a ) after viewpoint.

---

## Referee Comment (RC2) · C. Lemmen (Referee) · 19 May 2017

C. Lemmen (Referee)

carsten.lemmen@hzg.de

**General comments**

This manuscript by Pelupessy and colleagues introduces a new software tool that facilitates the integration of existing ocean modelling softwares into a modular and coupled simulation system and that facilitates the deployment of these coupled systems on different compute platforms. This software tool, called "OMUSE" builds on an existing set of tools that are used in astronomy ("AMUSE"), but adds to this wrapping codes for selected ocean models and a python user interface.

The manuscript is within the scope of Geoscientific Model Development. The work referred is highly relevant for the geoscientific research community. It is one of several current innovative efforts to facilitate collaboration between different ocean models; its

particular strengths are the modular design and the focus on usability.

The paper is overall well written, but suffers from jargon, some inconsistencies, carelessness in definitions, insufficient documentation for reproduction, all of which are detailed below, and which necessicate a thorough revision. Figures are appropriate and of good quality, but those containing code examples should be moved to an appendix.

There could be more specific comparisons with alternative approaches that would help the reader to make choices regarding the use of this or an alternative modeling environment. Also, the software must be deposited in a public and permanent repository and be marked with a persistent identifier such as a DOI.

I recommend that this paper is published pending these revisions.

**Specific comments**

**Title** The (revised) title and the text refer to OMUSE as a *framework*. This is, unfortunately, a term that us not clearly defined and that is overly used these days. It would better be avoided or clearly stated what its meaning is in the current context. I suggest to refer to OMUSE as a "toolset for coupling".

**Title and Abstract** I would find it useful to see the complete meaning of OMUSE as "Oceanographic Multipurpose Software Environment" reflected in the title. It is confusing to have it referred to as a "framework" vs. "software environment" and "multipurpose" vs. multimodel in the first line of the text and the title, respectively.

**Abstract** I don't see how OMUSE facilitates the design of coupled models. It certainly helps to implement the coupling. Reword more precisely what OMUSE does.

**l 21f** [1] Give full names and references at first occurrence of model acronyms. You might also consider to refer to a table where the models are listed with their institution
* * *
[1]Line numbers refer to the revised version that is available as a supplement to RC1 author comment

and reference.

**l 23f** These models are only a subset of the CMIP5 models, they don't constitute it.

**l 33** Many would not term SWAN a "coastal ocean model", it is a coastal wave model. Throughout the text, you could give the reader better structure for your coupling applications between (1) different ocean models (e.g. global to coastal), (2) ocean - wave coupling, and (3) data assimilation.

**l 42ff** There are many ways how couplings can be implemented, and many categories that describe the coupling type. Most often, the term "tight" versus "loose" coupling are used; in your case, the differentiation is between "monolithic" and "modular" coupling (and there are a lot of in-betweens). Refer to Valcke (2012) for categories. AGRIF is one tool to facilitate exchange of information that is on different grids. It may be helpful in both monolithic or modular approaches so it is not a useful category here. Then, of course, it is typically not sufficient (but already beyond many existing solutions) to couple just two different models (you propose coastal and open ocean) to tackle the coastal research problems you describe earlier.

**l 57ff** This sentence is only understandable to coupling experts. Please reword in more simple terms and explain jargon.

**l 68** Why do you restrict your work to the ocean modeling community? I believe your approach would be valuable for the entire Earth System modeling community and you should confidently state this, even though the applications presented are from the ocean domain.

**Figs 2,3,4,6** The code examples (provided in Figs 2,3,4,6) are not relevant to understanding the text and should be moved to an appendix. As they are, they are not easily understood and distract from the text.

Interactive
comment

[Figure]

**l 107ff** contains again too much jargon and should be rewritten for a broader audience. The possibility of running multiple instances of the same program, and even multiple instances that are compiled differently could be highlighted more, as this approach is an outstanding characteristic of OMUSE.

**l 118** "as the master script". You have not defined what this is.

**Section 2.1 overall** It would be helpful to have references and acronym lists and definitions (such as the table recommended before) for the technical terms MPI, OpenMPI, OpenMP, MPI-2. The (sometimes subtle) differences between these technologies are very likely unclear to much of the readership and must be explained as far as this distinction is relevant to the purpose of this paper. What about vendor-specific MPI implementations?

**l 181ff** Is there any use of the "particle" set in your applications? All ocean applications are defined on structured or unstructured grids ("meshes"). The "particle" concept seems useful for Lagrangian tracer studies and for observation data; I don't see any of your applications making use of it (or is the eddy tracker one of these?)

**l 732ff** As the need for coupling models is increasing and tackled by several new frameworks or toolkits/software environments please justify just how easy it would be to create more interoperability. Why an entirely new approach in OMUSE? Both ESMF (Hill et al. 2004) and CSMDS (Peckham 2013) et al. contain python interfaces; both support C and Fortran, and CSDMS many more language implementations. You already justify this a little with your state model and OO approach. But would it be not more helpful to provide BMI to ADCIRC/POP/SWAN ... (actually, SWAN already has one) such that these models can be used both in OMUSE and CSDMS? And to elaborate on existing BMI within OMUSE by wrapping the original BMI in your high-level OO?

**General discussion** I also think that you should contrast your work more to the most

recent efforts done in other coupling frameworks/software environments, such as the work in the GMD special issue by Balaji, or Cossarini's BFM coupler (also GMD), to name just a few. The purpose of this comparison would also be to highlight you own strengths and to give the reader more information on when not to choose your software but rather a different one.

**l 770f** this is wrong. GPL does not refer to private use in any way. Please convey the important terms of the GPL correctly. I would also find it helpful to include in the discussion a paragraph on your choice of license, i.e. GPL, as this choice imposes severe limitations (strong copyleft) on the distribution of coupled models.

**l 773f** It was not possible to download ADCIRC without registration; this should be stated (SWAN, POP worked). Also, with a serious attempt to build AMUSE and OMUSE according to the instructions provided, I was not able to achieve a successful build (though all requirements were met). While this may be a particular problem on my side (OSX + gfortran system, error in the python build script), it is not acceptable for what you call "production ready" software to not point the user to help/bug database/contact person. There is an "issue tracker" on the project site, but this seems to be inactive (only four issues, more than 8 months old).

**l 774f** Instructions on how to contribute are missing, particularly a contributor license agreement.

**General code availability** It is not clear what OMUSE v1.0 refers to. Please push your software to a permanent repository and obtain a DOI for for the published version (e.g. Zenodo). Bitbucket is a private company and cannot guarantee availability.

**Technical comments**

**l 16,291** Don't use "relatively" if no relation is provided.

**l 16f** Repetitious use of "also"

**l 36f** "relax" is jargon for physical modelers; try to address a general readership.

**l 43** Spelling of "AGRIFF", correct is AGRIF.

**l 68** Spelling "seperate", correct to "separate"

**l 118f** 118 relation of "its" and "it" unclear.

**l 138** misuse of "reckon"

**l 378** add "Eq." before (3)

---

## Author Response (AR1)

We have prepared a new version of the manuscript with the comments from the editor and the referees taken into account. Below we list the comments and our response to each point raised.

Comments from the executive editor
* * *
In order to simplify reference to your developments, please add a version number and consider to add the models acronym in the title of your article in your revised submission to GMD.

   RESPONSE: We have changed the title.

Additionally, I like to point out, that the Code Availability and the Acknowledge- ment Sections are distinctive parts of the overall paper structure (see the section "manuscript composition" on http://www.geoscientific-model-development.net/for_authors/ manuscript_preparation.html). In contrast to this, "Extending OMUSE" is not part of this overall structure. Therefore this section is in the wrong place. It should be either a section in the main body of the paper or an appendix section, but it should definitely not be placed between Code Availability section and Acknowledgements .

   RESPONSE: We have moved the section "Extending OMUSE" to be subsection 3.4.

Comments from anonymous referee #1
* * *
- Title: Please provide more details about OMUSE in the title (not just its name) so to better reflect the content of the paper. Also, it is recommended to provide in the title a version number of the latest OMUSE version available (see also A. Kerweg's comments).

   RESPONSE: We have changed the title to conform to journal guidelines.

- L15: I am not sure the application examples described show the efficiency of OMUSE; I would insist more on the flexibility but less on the efficiency.

   RESPONSE: We have have removed "efficient."

- L63: You write "This has the benefit of the parallelism and . . ." . This seems incoherent to me with what you write in the discussion L716-L717: "limitation of the current design of OMUSE is the fact that the communication between solvers is handled by the master script. This imposes a bottleneck for the performance of the communication between e.g. two parallel codes." Also, L105: What is the benefit of providing built-in parallelism in the MPI-based remote function protocol if, as written in the discussion the handling of the communication through the master script imposes a bottleneck? Can you clarify?

   RESPONSE: The current setup provides for independent and parallel running different codes, while the current communication implementation still has a bottleneck (which can be removed). We have added clarification and a reference to the discussion (in section 6) to the description of the interface in section 2.1.

- L64 and other places: You mention the "bookkeeping inherent to code coupling". Can you explain what you mean by bookkeeping? In particular, what do you mean precisely L441-442 by "extensive automation of bookkeeping operations"?

   RESPONSE: We agree that its use was not clear. We explain the meaning of bookkeeping in section 2.4 (Data model) now, and clarify it in section 4. The "extensive automation of bookkeeping operations" has been rephrased.

- L116: What do you mean by "the sockets channel is mainly useful for cases were a component process is to be run on one machine". Of course,

a process is always run on a machine! Do you mean "the sockets channel is mainly useful for cases where a component process is to be run a different machine".

  RESPONSE: In the current setup, the sockets communication is mainly used for inter process communication within a machine. This is clarified now.

– L126: you write: "the communication requirements between processes must not be too demanding. Where this is not the case (e.g. when a strong algorithmic coupling is necessary) a different approach may be more appropriate." Why should the (MPI) communication be "not too demanding"? What do you mean by "a different approach"? Do you mean something else that OMUSE?

  RESPONSE: "Demanding" refers to the amount of communication necessary between processes. We have removed these sentences ("Additionally ... more appropiate"), since this refers more too the communication bottlenecks discussed later on.

– L190–191: I am not sure I understand the difference between Cartesian and Regular. What do you mean by "constant"? constant in time (I suppose not)? Constant with respect to the dimension? Please clarify.

  RESPONSE: Cartesian = same constant cellsize in each dimension, Regular = constant cellsize in each dimension, potentially different for each dimension. We have clarified this in the text.

– L241: can you clarify what you mean by "This functionality is preferably not used within OMUSE." Does OMUSE support code using their own I/O library or not? The word "preferably" is ambiguous.

  RESPONSE: We now explicitly state that use of the original I/O is supported (and give a potential reason to do so).

– L385–388: What is the relation between these two sentences (starting with "In case of stationary..." ) and the restrictions on the forcings discussed in this paragraph?

  RESPONSE: None. A new paragraph is started now.

– L401–402: SCRIP library is included in OASIS but it is not in MCT, even if MCT can use grid remapping weights and addresses generated (separately) by SCRIP.

  RESPONSE: We have clarified the text to make this distinction.

– L422–423: the source grid has to be structured not because of the way SCRIP com– putes the area integrals but for the calculation of the gradients needed for the 2nd order.

  RESPONSE: We have changed this in the text.

– L486: Section 4.1.2 is about non-overlapping domains. Why is it named "Domain decomposition"? Also, are you talking here really about "non-overlapping" domains or more precisely "partially-overlapping" domains, as in the 5.2 example?

  RESPONSE: The solution is obtained on the union of overlapping domains, so its a matter of viewpoint whether to call this "Domain decomposition." We have thought about another heading to this subsection, but we prefer the current one as the most concise. The domains can be non-overlapping in so far a small overlap is necessary for the boundary conditions on the respective domains. We have added "partially overlapping" in parentheses.

– L507: What do you mean by "preceding examples"? Are these the 4.1.1 and 4.1.2 couplings? If so, maybe put the numbers 4.1.1 and 4.1.2 for clarity.

  RESPONSE: added the section numbers.

Minor comments:
- L14: Remove the , after "solver"

   RESPONSE: fixed

- L23: I would not write "current CMIP5" as CMIP5 is over now and CMIP6
is on-going.

   RESPONSE: fixed (removed "current")

- L123: maybe replace "the requested subroutine calls" by "the requested
simulation code subroutine calls" (if I got this right)?

   RESPONSE: yes, implemented suggestion.

- L144, change "OMUSE implements a" for "OMUSE implements also a"

   RESPONSE: done

- L509, add a ) after viewpoint.

   RESPONSE: fixed

Comments from referee #2
* * *
- Title: The (revised) title and the text refer to OMUSE as a framework.
This is, unfortunately, a term that us not clearly defined and that is
overly used these days. It would better be avoided or clearly stated
what its meaning is in the current context. I suggest to refer to OMUSE
as a "toolset for coupling".

  RESPONSE: Although we agree the term framework is somewhat nebulous,
  it is often used for a set of components (tools, APIs etc) that can be
  used to construct an application in a particular domain (where the
  flow of the application is more strongly prescribed than when using
  e.g. a library). We have dropped the term from the title (see below)
  and from the abstract. In the introduction we have added a short
  footnote with an explanation of the term framework as used here.

- Title and Abstract: I would find it useful to see the complete meaning
of OMUSE as "Oceanographic Multipurpose Software Environment" reflected
in the title. It is confusing to have it referred to as a "framework" vs.
"software environment" and "multipurpose" vs multimodel in the first
line of the text and the title, respectively.

  RESPONSE: we have adapted the title, retaining the acronym and
  version number.

- Abstract: I don't see how OMUSE facilitates the design of coupled
models. It certainly helps to implement the coupling. Reword more
precisely what OMUSE does.

  RESPONSE: We have added a short sentence to the abstract describing
  the facilities provided by OMUSE that aid model coupling (and
  slightly rephrased the abstract to accomodate this).

- l 21f: Give full names and references at first occurrence of model
acronyms. You might also consider to refer to a table where the models
are listed with their institution

  RESPONSE: We give the full name of the models now. Here and below we
  have decided against putting all acronyms in a table, since most of
  them are referred to only once or twice (and not really needed to
  understand the main text).

- l 23f: These models are only a subset of the CMIP5 models, they don't
constitute it.

  RESPONSE: changed "constitute" -> "can be used as"

- l 33: Many would not term SWAN a "coastal ocean model", it is a
coastal wave model. Throughout the text, you could give the reader
better structure for your coupling applications between (1) different
ocean models (e.g. global to coastal), (2) ocean - wave coupling, and
(3) data assimilation.

  RESPONSE: changed "regional coastal ocean models" to "regional models"

- l 42ff: There are many ways how couplings can be implemented, and many
categories that describe the coupling type. Most often, the term "tight"
versus "loose" coupling are used; in your case, the differentiation is
between "monolithic" and "modular" coupling (and there are a lot of
in-betweens). Refer to Valcke (2012) for categories. AGRIF is one tool
to facilitate exchange of information that is on different grids. It
may be helpful in both monolithic or modular approaches so it is not a
useful category here. Then, of course, it is typically not sufficient
(but already beyond many existing solutions) to couple just two
different models (you propose coastal and open ocean) to tackle the
coastal research problems you describe earlier.

  RESPONSE: We have adapted these lines according to the suggestions (
  we contrast monolithic and modular approaches now).

– l 57ff: This sentence is only understandable to coupling experts.
Please reword in more simple terms and explain jargon.

  RESPONSE: We have rewritten this sentence, more carefully explaining
  the difference between integrated and library approaches.

– l 68: Why do you restrict your work to the ocean modeling community? I
believe your approach would be valuable for the entire Earth System
modeling community and you should confidently state this, even though
the applications presented are from the ocean domain.

  RESPONSE: Indeed, OMUSE can be used more widely (and it is already
  being used for e.g. coupling meteorological models). We have rephrased
  this sentence to reflect this.

– Figs 2,3,4,6: The code examples (provided in Figs 2,3,4,6) are not
relevant to understanding the text and should be moved to an appendix.
As they are, they are not easily understood and distract from the text.

  RESPONSE:  We have often found that the code papers describing
  coupling tools leave out this kind of information, which makes it
  difficult to get a grasp of the practical use of such tools and we
  think it helps the reader get an idea about this by providing some
  definitive examples. Admittedly, this is subjective so if the editor
  thinks it is better to move these (also maybe for layout reasons), we
  are happy to act on this.

– l 107ff: contains again too much jargon and should be rewritten for a
broader audience. The possibility of running multiple instances of the
same program, and even multiple instances that are compiled
differently could be highlighted more, as this approach is an
outstanding characteristic of OMUSE.

  RESPONSE: We have rewritten this and subsequent sentences following
  this paragraph to be more accessible.

– l 118: "as the master script". You have not defined what this is.

  RESPONSE: We have replaced "master script" with "user script" (which
  was introduced at the beginning of section 2).

– Section 2.1: overall It would be helpful to have references and
acronym lists and definitions (such as the table recommended before)
for the technical terms MPI, OpenMPI, OpenMP, MPI-2. The (sometimes
subtle) differences between these technologies are very likely unclear
to much of the readership and must be explained as far as this
distinction is relevant to the purpose of this paper. What about
vendor-specific MPI implementations?

  RESPONSE: We have added the acronyms at the first use of MPI (2.1) and
  a footnote for OpenMP (end of 2.1). Vendor specific implementations of
  MPI can also be used, this is now briefly mentioned (2.1).

– l 181ff: Is there any use of the "particle" set in your applications?
All ocean applications are defined on structured or unstructured grids
("meshes"). The "particle" concept seems useful for Lagrangian tracer
studies and for observation data; I don't see any of your applications
making use of it (or is the eddy tracker one of these?)

  RESPONSE: Most ocean codes use the grid data structures. Indeed, a
  Lagrangian tracker could use them (the eddy tracker can also use
  particle sets). We mention possible uses of the particle sets for
  ocean applications now in 2.4.

– l 732ff: As the need for coupling models is increasing and tackled by
several new frame- works or toolkits/software environments please
justify just how easy it would be to create more interoperability. Why
an entirely new approach in OMUSE? Both ESMF (Hill et al. 2004) and
CSMDS (Peckham 2013) et al. contain python interfaces; both support C
and Fortran, and CSDMS many more language implementations. You already
justify this a little with your state model and OO approach. But would

it be not more helpful to provide BMI to ADCIRC/POP/SWAN ... (actually, SWAN already has one) such that these models can be used both in OMUSE and CSDMS? And to elaborate on existing BMI within OMUSE by wrapping the original BMI in your high-level OO?

   RESPONSE: We think the discussion of interoperability makes most sense in relation to the CSDM (since the specification of the BMI translates more easily into an OMUSE interface than e.g. the specification of ESMF interfaces). We have expanded the discussion on the differences of CSDM and OMUSE and their interoperability.

– General discussion: I also think that you should contrast your work more to the most recent efforts done in other coupling frameworks/software environments, such as the work in the GMD special issue by Balaji, or Cossarini's BFM coupler (also GMD), to name just a few. The purpose of this comparison would also be to highlight you own strengths and to give the reader more information on when not to choose your software but rather a different one.

   RESPONSE: We have added more discussion contrasting OMUSE with other coupling frameworks.

– l 770f: this is wrong. GPL does not refer to private use in any way. Please convey the important terms of the GPL correctly. I would also find it helpful to include in the discussion a paragraph on your choice of license, i.e. GPL, as this choice imposes severe limitations (strong copyleft) on the distribution of coupled models.

   RESPONSE: The licensing comment was incorrectly associated with private use. This is clearly separated now. In response to concerns (and together with similar discussions within the AMUSE community) we have changed the license to the more permissive Apache 2.0. This is mentioned (and briefly discussed) in the paper now.

– l 773f: It was not possible to download ADCIRC without registration; this should be stated (SWAN, POP worked). Also, with a serious attempt to build AMUSE and OMUSE according to the instructions provided, I was not able to achieve a successful build (though all requirements were met). While this may be a particular problem on my side (OSX + gfortran system, error in the python build script), it is not acceptable for what you call "production ready" software to not point the user to help/bug database/contact person. There is an "issue tracker" on the project site, but this seems to be inactive (only four issues, more than 8 months old).

   RESPONSE: We did not want to mention ADCIRC explicitly since its licensing may change in the future (as for as we understand they are planning to become fully open source). In order to alleviate problems with the build system (especially for cases were people want to test things out) we will make available precompiled binaries on the OMUSE project site. The issue tracker is actively monitored, but indeed most installation issues are handled through the AMUSE issue tracker (which is much more active).

– l 774f: Instructions on how to contribute are missing, particularly a contributor license agreement.

   RESPONSE: We have added a short comment on contributions. We do not think at this stage a formal contributor license policy is necessary (very few scientific projects of this size have one), but we will monitor prevailing practices and implement one when necessary.

– General code availability: It is not clear what OMUSE v1.0 refers to. Please push your software to a permanent repository and obtain a DOI for for the published version (e.g. Zenodo). Bitbucket is a private company and cannot guarantee availability.

   RESPONSE: we have made available a tagged snapshot, which is labelled v1.0 on the project website and available on the Zenodo archive.

Technical comments
* * *
– l 16,291 Don't use "relatively" if no relation is provided.

   RESPONSE: removed both occurences of "relatively"

– l 16f Repetitious use of "also"

   RESPONSE: fixed (dropped both)

– l 36f "relax" is jargon for physical modelers; try to address a general readership.

   RESPONSE: we have rephrased this

– l 43 Spelling of "AGRIFF", correct is AGRIF.

   RESPONSE: corrected

– l 68 Spelling "seperate", correct to "separate"

   RESPONSE: corrected

– l 118f 118 relation of "its" and "it" unclear.

   RESPONSE: fixed

– l 138 misuse of "reckon"

   RESPONSE: fixed

– l 378 add "Eq." before (3)

   RESPONSE: fixed

[revised manuscript text omitted]

---

## Author Response (AR2)

Below our replies to the minor points raised by the referee:

Within the new contextualization section, the authors write ''The similarities of the CSDMS and OMUSE interfaces translates, in principle, into interoperability ... the interface components of a code in the CSDMS can be converted to an OMUSE interface.....'' It is be beyond the scope of the current manuscript, but I suggest that the authors reach out the CSDMS (par exemplum) community and develop and publish such an interface as an effort to bridging existing technologies.

REPLY: We thank the referee for the suggestions; indeed, this is one of the things we are looking into!

Technical corrections:

– The changes in the manuscript (visible in track changes mode) need thorough spell-checking

REPLY: we have checked the text, and made corrections (also some minor rewording)

– Note the revised notation for dois, e.g. "https://doi.org/" instead of "doi:"

REPLY: corrected

[revised manuscript text omitted]